# Reward and aversion processing by input-defined parallel nucleus accumbens circuits in mice

Kuikui Zhou[1,2,9], Hua Xu[1,9], Shanshan Lu[1,3,9], Shaolei Jiang[1,4], Guoqiang Hou [1], Xiaofei Deng[1], Miao He [5] & Yingjie Zhu [1,3,6,7,8] ✉

The nucleus accumbens (NAc) is critical in mediating reward seeking and is also involved in negative emotion processing, but the cellular and circuitry mechanisms underlying such opposing behaviors remain elusive. Here, using the recently developed AAV1-mediated anterograde transsynaptic tagging technique in mice, we show that NAc neurons receiving basolateral amygdala inputs (NAc[BLA]) promote positive reinforcement via disinhibiting dopamine neurons in the ventral tegmental area (VTA). In contrast, NAc neurons receiving paraventricular thalamic inputs (NAc[PVT]) innervate GABAergic neurons in the lateral hypothalamus (LH) and mediate aversion. Silencing the synaptic output of NAc[BLA] neurons impairs reward seeking behavior, while silencing of NAc[PVT] or NAc[PVT]→LH pathway abolishes aversive symptoms of opiate withdrawal. Our results elucidate the afferent-specific circuit architecture of the NAc in controlling reward and aversion.

Reward-seeking and threat avoidance are critical for survival, and the nucleus accumbens (NAc) is involved in orchestrating both processes. The NAc plays an important role in regulating drug reward, feeding, social interaction, pain, and instrumental learning[1–4]. In addition, NAc dysfunction has been implicated in anxiety, depression, anhedonia, and substance addiction[5–7]. However, the functional and organizational principles of NAc in mediating positive and negative motivational valence remain largely unknown.

Despite the fact that medium spiny neurons (MSNs) in the NAc are highly heterogeneous[8], the prevailing model posits that dopamine receptor 1-expressing MSNs (D1R-MSN) and dopamine receptor 2-expressing MSNs (D2R-MSN) operate in opposite ways[9,10]. D1R-MSNs project directly to the VTA and were thought to control reward-seeking behavior, while D2R-MSNs relay signals to the VTA via the ventral pallidum (VP) and were supposed to contribute to aversion[11,12]. However, recent studies indicate that D1-MSNs also comprise a significant portion of the classical indirect pathway by synapsing on VP neurons that project to the VTA and are activated by aversive stimuli[13,14]. On the contrary, activation of D2R-MSNs could also drive reinforcement[15–17]. Therefore, this classic striatal direct and indirect pathway model is inappropriate when applied to the NAc[18,19].

Accumulating evidence based on optogenetic-assisted circuit dissection suggests that specific NAc pathways might be involved in different functions[20–22]. Studies have revealed that glutamatergic

[1]Shenzhen Key Laboratory of Drug Addiction, Shenzhen Neher Neural Plasticity Laboratory, the Brain Cognition and Brain Disease Institute, Shenzhen Institute of Advanced Technology, Chinese Academy of Sciences; Shenzhen-Hong Kong Institute of Brain Science-Shenzhen Fundamental Research Institutions, 518055 Shenzhen, China. [2]School of Health and Life Sciences, University of Health and Rehabilitation Sciences, 266071 Qingdao, China. [3]University of Chinese Academy of Sciences, 100049 Beijing, China. [4]University of Shanghai for Science and Technology, 200093 Shanghai, China. [5]Institutes of Brain Science, Department of Neurology, State Key Laboratory of Medical Neurobiology and MOE Frontiers Center for Brain Science, Zhongshan Hospital, Fudan University, 200032 Shanghai, China. [6]Faculty of Life and Health Sciences, Shenzhen Institute of Advanced Technology, Chinese Academy of Sciences, 518055 Shenzhen, China. [7]CAS Key Laboratory of Brain Connectome and Manipulation, the Brain Cognition and Brain Disease Institute (BCBDI), Shenzhen Institute of Advanced Technology (SIAT), Chinese Academy of Sciences, 518055 Shenzhen, China. [8]CAS Center for Excellence in Brain Science and Intelligence Technology, Chinese Academy of Sciences, 200031 Shanghai, China. [9]These authors contributed equally: Kuikui Zhou, Hua Xu, Shanshan Lu. ✉e-mail: yj.zhu1@siat.ac.cn

transmission from the thalamic paraventricular nucleus (PVT) to the NAc drives aversion[6,23], whereas canonical glutamatergic inputs, such as that from the basolateral amygdala (BLA) have been linked to reward processing[24,25]. It remains a mystery why distinct glutamatergic inputs to the NAc produce opposite behavioral consequences. Thus, we sought to investigate whether the PVT and the BLA inputs define a separation of NAc subcircuits that process positive and negative motivational valence, respectively.

## Results

### Transsynaptic tagging of neurons receiving input from the BLA or the PVT

In this study, we took advantage of recently developed AAV1-mediated anterograde transsynaptic tagging[26–28], to label NAc neurons that are innervated by specific afferents. We first injected AAV1-Cre into the BLA or PVT of Ai14 (Cre-dependent tdTomato reporter) mice and examined the labeled neurons throughout the brain. In the mice with BLA injection, tdTomato-expressing cell bodies were observed in major regions known to be directly targeted by BLA[29,30], including the medial prefrontal cortex (mPFC), the NAc, the bed nucleus of the stria terminalis (BNST), and the ventral hippocampus (vHipp) (Supplementary Fig. 1a). In the mice with PVT injection, tdTomato-positive cell bodies were observed in mPFC, NAc, BNST, and the central nucleus of the amygdala (CeA) (Supplementary Fig. 1b), which is reminiscent of PVT's projection pattern[31,32]. To verify the monosynaptic connection, we expressed ChR2 in the PVT or BLA, and performed targeted patch-clamp recording from tdTomato-positive neurons in NAc slices (Supplementary Fig. 1c). In labeled cells, blue light evoked robust excitatory postsynaptic currents (EPSCs) and inhibitory postsynaptic currents (IPSCs) (Supplementary Fig. 1d, e). The EPSCs were abolished in the presence of CNQX, while they remained in the presence of TTX and 4-AP[33], suggesting the monosynaptic glutamatergic transmission (Supplementary Fig. 1d, e). The IPSCs exhibited longer latency than that of EPSCs, and they were abolished by CNQX application or TTX & 4-AP application, suggesting disynaptic feed-forward inhibition (Supplementary Fig. 1d, e). These results demonstrate the utility of the AAV1-mediated transsynaptic tagging strategy in the labeling of input-defined NAc neuronal populations.

### Segregated distribution of NAc[BLA] and NAc[PVT] neurons

To examine the potential overlap between NAc neurons innervated by BLA and PVT inputs, we produced Cre/Flp double-reporter mice by crossing R26R-EYFP mice (Cre-dependent EYFP reporter line)[34] and FSF-tdTomato mice (Flp-dependent tdTomato reporter line)[35]. In Cre/Flp double-reporter mice, we injected AAV1-Cre into the PVT and AAV1-Flp into the BLA. NAc neurons innervated by the BLA (NAc[BLA]) were labeled with tdTomato, while PVT innervating NAc neurons (NAc[PVT]) were EYFP-positive (Fig. 1a, b). NAc[BLA] neurons were distributed throughout the core and shell of NAc subregions (Fig. 1a and Supplementary Fig. 2), which is consistent with the distribution of BLA axons in the NAc reported in previous studies[24,25]. NAc[PVT] neurons were also distributed throughout the NAc subregions and more densely in the medial shell of NAc[6] (Fig. 1a and Supplementary Fig. 2). Interestingly, tdTomato-positive NAc[BLA] neurons and EYFP-positive NAc[PVT] neurons were largely segregated, with only a small portion of neurons positive for both tdTomato and EYFP (12.8% for tdTomato+ and 9.0% for EYFP+ neurons) (Fig. 1b, c). Next, we examined whether input-defined NAc neurons specifically express certain types of dopamine receptors. We performed in situ hybridization of dopamine receptor 1 (D1R) and dopamine receptor 2 (D2R) on NAc slices, and examined the overlap of D1R and D2R with NAc[BLA] and NAc[PVT] neurons (Fig. 1d, f). Interestingly, we found the percentage of D1R or D2R expression is not different between NAc[BLA] and NAc[PVT] neurons (Fig. 1e, g). Therefore, our results suggest functional differences between the BLA→NAc and PVT→NAc pathways do not depend on the type of dopamine receptor expression

in NAc[BLA] and NAc[PVT] neurons, pending a full investigation of the output pattern of these two subpopulations.

### Activation of NAc[BLA] and NAc[PVT] neurons induces reinforcement and aversion, respectively

Before further analyzing the input-output relationships of these two subtypes of neurons labeled with the AAV1-mediated transsynaptic tagging method, we first examined whether NAc[BLA] and NAc[PVT] neurons exhibit functional divergence and differently contribute to positive and negative motivational valence. To selectively activate NAc[BLA] or NAc[PVT] neurons, we injected AAV1-Cre to the BLA or PVT and AAV-DIO-ChR2 into the NAc of wild-type mice (Fig. 2a, d). Mice expressing ChR2 in NAc[BLA] neurons readily learned to perform nose-poke to earn optical stimulation, while mice expressing ChR2 in NAc[PVT] neurons did not (Fig. 2b, c). In the real-time place preference (RTPP) test, optogenetic activation of NAc[BLA] neurons increased the time spent in the chamber paired with light stimulation, whereas activation of NAc[PVT] neurons reduced the time spent in the light-paired chamber in both male and female animals (Fig. 2e, f and Supplementary Fig 6). These results suggest that NAc[BLA] and NAc[PVT] neurons mediate opposite motivational valence.

### Different contributions of NAc[BLA] and NAc[PVT] neurons in appetitive and aversive responses

To further examine the physiological role of NAc[BLA] and NAc[PVT] neurons in appetitive and aversive behaviors, we transduced those neurons with tetanus neurotoxin (TeNT) for synaptic silencing (Fig. 2g, j). Our electrophysiological data showed that the expression of TeNT is very efficient in blocking synaptic transmission (Supplementary Fig. 1f–h). Silencing the NAc[BLA] neurons with TeNT significantly reduced animals' motivation to seek palatable food (Ensure), but did not affect the aversive physical responses of naloxone-precipitated opioid withdrawal (Fig. 2i, l). On the contrary, silencing NAc[PVT] neurons dramatically reduced the physical signs of opioid withdrawal, but had no effect on food-seeking behavior (Fig. 2i, l). Our data support the functional difference between the BLA→NAc and PVT→NAc pathways, and suggest that afferent-specific NAc subpopulations exert opposite roles in motivated behaviors.

### Different projection profiles of NAc[BLA] and NAc[PVT] neurons

We investigated whether NAc[BLA] and NAc[PVT] neurons might project to different downstream brain areas to elicit distinct behaviors. We systematically examined the whole brain axonal projection of NAc[BLA] and NAc[PVT] neurons, by injecting AAV1-Cre into the BLA or the PVT, and AAV-DIO-GFP into the NAc of wild-type mice. We found that NAc[BLA] neurons mainly project to the VP, the lateral hypothalamus (LH), and the VTA[36]. Although NAc[PVT] neurons also project to the VP and the LH, the axonal projections to the VTA are very sparse (Fig. 3a, b). These results suggest that the VTA received inputs from NAc[BLA] neurons but not NAc[PVT] neurons. To further confirm this connectivity scheme, we labeled VTA-projecting NAc neurons with retrograde tracer CTB-488 and LH-projecting NAc neurons with CTB-647. In the same mouse, we also labeled NAc[BLA] or NAc[PVT] neurons with AAV1-Cre strategy (Supplementary Fig. 3a). Careful examination revealed that the overlap ratio between LH-projecting NAc neurons and NAc[BLA] was as high as the overlap between LH-projecting NAc neurons and NAc[PVT] neurons. However, the overlap between VTA-projecting NAc neurons and NAc[PVT] neurons was significantly less than the overlap between VTA-projecting NAc neurons and NAc[BLA] neurons (Supplementary Fig. 3b, c). These results confirm that although both NAc[BLA] and NAc[PVT] neurons project to LH, only NAc[BLA] neurons project to VTA. To further demonstrate these specific functional connectivities, we transduced NAc[BLA] or NAc[PVT] neurons with ChR2 and performed patch-clamp recording in VTA slices (Fig. 3c). Light stimulation of NAc[BLA] axon terminals in the VTA slice evoked picrotoxin-sensitive inhibitory

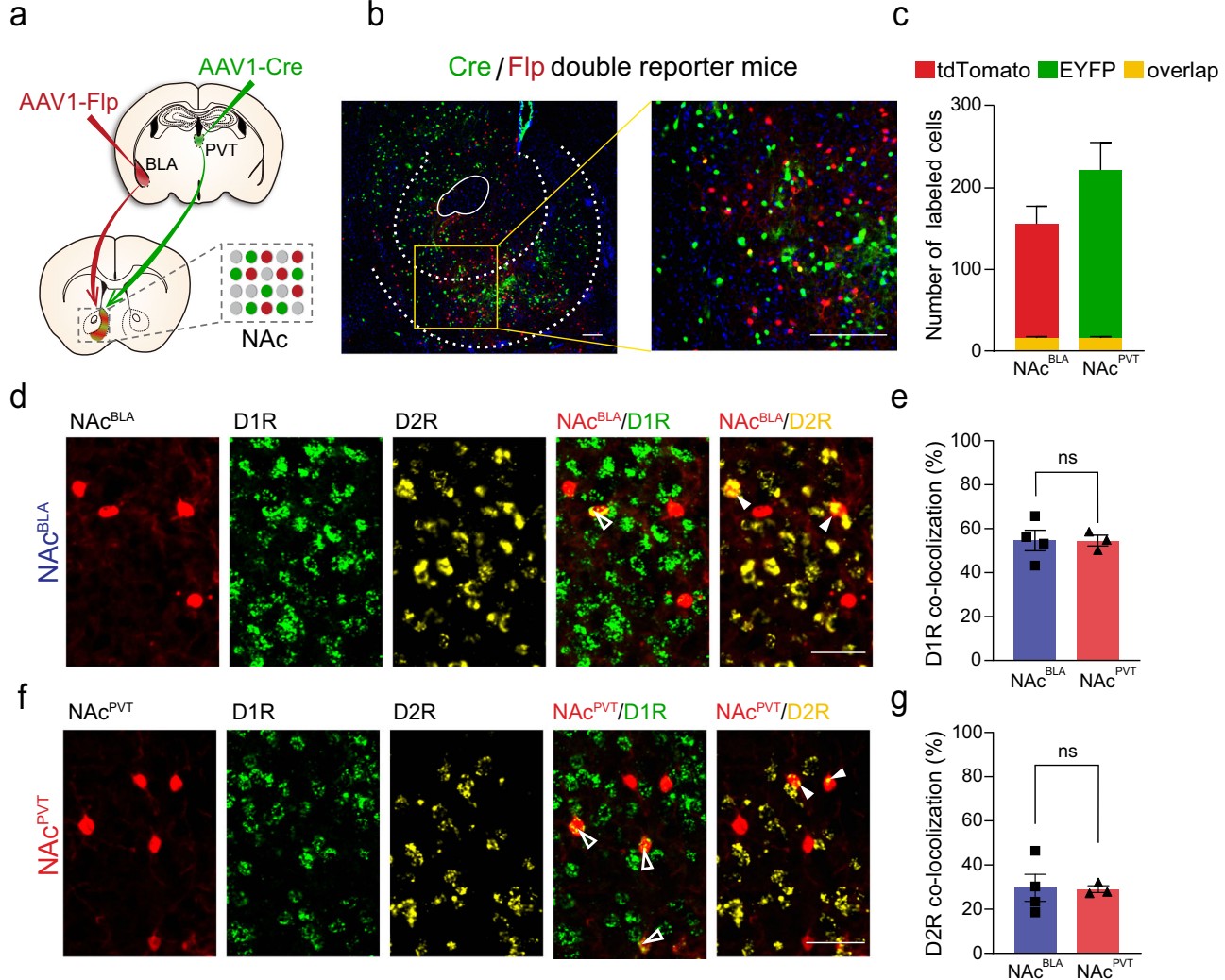

**Fig. 1 | Distribution of NAc^BLA and NAc^PVT neurons in the NAc. a** Strategy using AAV1-mediated anterograde transsynaptic tagging to label NAc neurons receiving BLA input (NAc^BLA) and NAc neurons receiving PVT input (NAc^PVT) in Cre/Flp double-reporter mice. **b** Example image showing labeled NAc^BLA (red) and NAc^PVT (green) neurons. Scale bar: 100 μm. **c** Quantification of the number of anterograde labeled NAc^BLA and NAc^PVT neurons in the NAc in Cre/Flp double-reporter mice (n = 5). Mean ± s.e.m. **d** Example images showing the distribution of NAc^BLA neurons (red), D1R-MSNs (green), and D2R-MSNs (yellow) in NAc.

Arrowheads indicate co-labeled cells. Scale bar: 50 μm. **e** Quantification of the percentage of NAc^BLA neurons (n = 4 slices) and NAc^PVT neurons (n = 3 slices) expressing D1R. Two-tailed Mann–Whitney test. ns not significant. Mean ± s.e.m. **f** Example images showing the distribution of NAc^PVT neurons (red), D1R-MSNs (green), and D2R-MSNs (yellow) in NAc. Arrowheads indicate co-labeled cells. Scale bar: 50 μm. **g** Quantification of the percentage of NAc^BLA neurons (n = 4 slices) and NAc^PVT neurons (n = 3 slices) expressing D2R. Two-tailed Mann–Whitney test. ns not significant. Mean ± s.e.m.

postsynaptic current (IPSC) in 57.9% (11/19) of VTA neurons, while stimulation of NAc^PVT axon terminals only evoked IPSC in 10% (2/20) of VTA neurons (Fig. 3d, e). Among those responsive VTA neurons, the average amplitude of light-evoked IPSCs was much smaller when stimulating the NAc^PVT→VTA pathway, compared to that of the NAc^BLA→VTA pathway (Fig. 3f).

### The rewarding effect of NAc^BLA→VTA activation depends on dopamine signaling

Since we found the NAc^BLA neurons form stronger connections with the VTA than NAc^PVT neurons do, we compared the behavioral outcomes of activating the NAc^BLA→VTA and NAc^PVT→VTA pathway (Fig. 4a). When directly activating the NAc^BLA→VTA pathway, we found that mice readily learn optical self-stimulation and increase the time spent in the light-paired chamber in RTPP test (Fig. 4b–d). Activating the NAc^PVT→VTA pathway did not produce similar behavioral outcomes (Fig. 4b–d). We reasoned the rewarding effect of optical activation of the NAc^BLA→VTA

pathway might act through the disinhibition of dopaminergic neurons[37]. To test this possibility, we activated NAc^BLA terminals in the VTA, and simultaneously recorded calcium activity of GABAergic neurons and dopaminergic neurons in the VTA using GAD2-Cre and DAT-Cre mice, respectively (Fig. 5a). Optogenetic excitation of the NAc^BLA→VTA pathway reduced the activity of GABAergic neurons (Fig. 5b), while the same stimulation evoked robust excitatory responses in dopaminergic neurons in the VTA (Fig. 5c). Next, we examined whether disinhibition of dopaminergic neurons increases dopamine release into the NAc. We recorded dopamine signals by expressing dopamine sensor GRAB_DA2m[38] in the NAc and delivered optogenetic stimulation to the NAc^BLA→VTA pathway (Fig. 5d). We found dopamine release in the NAc escalated upon increases in the stimulation frequency (Fig. 5e). To determine whether dopamine release is responsible for the rewarding effect of NAc^BLA→VTA activation, we examined the animals in the RTPP test after systematic injections of D1R or D2R antagonist. D1R antagonist (SCH-23390) completely abolished the rewarding effect of NAc^BLA→VTA

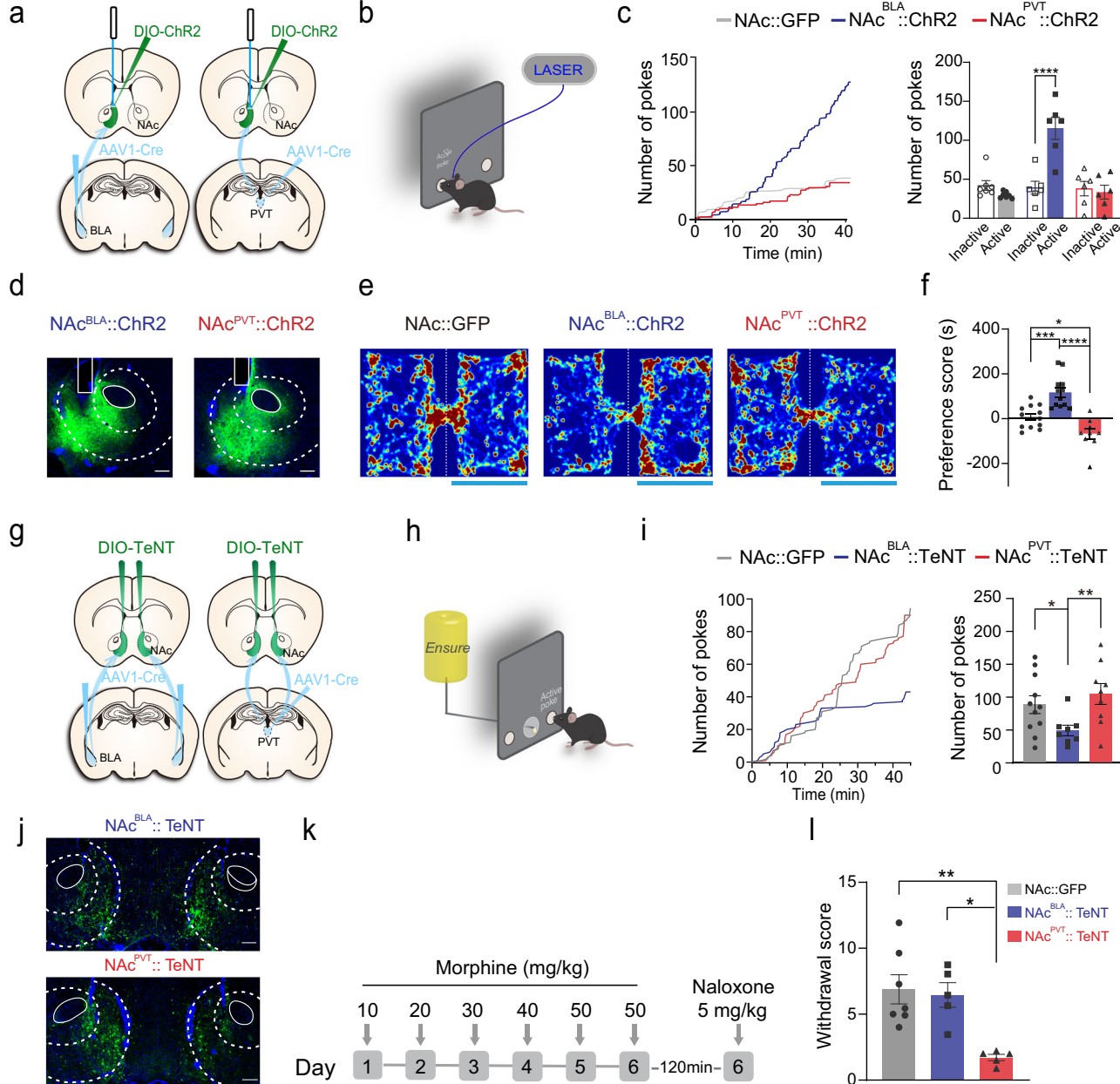

**Fig. 2 | Functional divergence of NAc^BLA and NAc^PVT subpopulations. a** Schematic showing the viral strategy to activate NAc^BLA or NAc^PVT neurons. **b** Schematic illustrating optogenetic self-stimulation task. **c** Left: example cumulative curves of the number of active nose-pokes made in 40 min behavioral sessions (FR = 1). Right: average numbers of nose-pokes for NAc::GFP (*n* = 7), NAc^BLA::ChR2 (*n* = 6) and NAc^PVT::ChR2 mice (*n* = 6). Two-way ANOVA: ChR2 × poke ($F_{(2,32)}$ = 16.27, *P* < 0.0001), ChR2 ($F_{(2,32)}$ = 15.41, *P* < 0.0001), followed by post hoc Sidak's test. *****P* < 0.0001. Mean ± s.e.m. **d** Examples of viral expression and locations of optic fibers. Scale bar: 200 μm. **e** Representative heatmaps for real-time place preference test. Left: non-laser paired side; Right: laser paired side. **f** Quantification of preference score in real-time place preference test for NAc::GFP (*n* = 12), NAc^BLA::ChR2 (*n* = 12) and NAc^PVT::ChR2 mice (*n* = 9). One-way ANOVA ($F_{(2,30)}$ = 21.94, *P* < 0.0001) followed by post hoc Tukey's test (NAc^BLA::ChR2 vs. NAc::GFP: *P* = 0.0007; NAc^PVT::ChR2 vs. NAc::GFP: *P* = 0.0351; NAc^BLA::ChR2 vs. NAc^PVT::ChR2: *P* < 0.0001).

Mean ± s.e.m. **g** Schematic showing the viral strategy to silence synaptic outputs of NAc^BLA and NAc^PVT neurons with tetanus toxin (TeNT). **h** Illustration of the palatable food-seeking task. **i** Left: example of cumulative curves of the number of active nose-pokes made to obtain palatable food reward in 45 min behavioral sessions (FR = 1). Right: number of nose-pokes for NAc::GFP (*n* = 11), NAc^BLA::TeNT (*n* = 8) and NAc^PVT::TeNT (*n* = 9) mice. One-way ANOVA ($F_{(2,25)}$ = 4.102, *P* = 0.0288) followed by post hoc Fisher's LSD test (NAc^BLA::TeNT vs. NAc::GFP, *P* = 0.0492; NAc^BLA::TeNT vs. NAc^PVT::TeNT, *P* = 0.0098). Mean ± s.e.m. **j** Examples of TeNT-EGFP expression. Scale bar: 200 μm. **k** Protocol for naloxone-induced morphine withdrawal. **l** Global morphine withdrawal scores for NAc::GFP (*n* = 7), NAc^BLA::TeNT (*n* = 5) and NAc^PVT::TeNT (*n* = 5) mice. One-way ANOVA ($F_{(2,14)}$ = 8.914, *P* < 0.01) followed by post hoc Tukey's test (NAc^PVT::TeNT vs. NAc::GFP, *P* = 0.0038; NAc^PVT::TeNT vs. NAc^BLA::TeNT, *P* = 0.0120). Mean ± s.e.m.

stimulation (Fig. 5f, h), while the D2R antagonist (Raclopride) had no effect (Fig. 5h, i).

**The aversive effect of NAc^PVT activation is mediated by the LH**

The VP and the LH are two major targets of NAc^PVT neurons (Fig. 3a), thus we probed the role of each downstream pathway independently.

Optogenetic activation of the NAc^PVT→LH evoked behavior avoidance, while activation of the NAc^PVT→VP projection had no effect in the RTPP test (Fig. 6a–c). Furthermore, activation of the NAc^PVT→LH pathway readily reduces palatable food intake (Fig. 6d–f). These results suggest that the LH might be the key downstream effector of NAc^PVT neurons for aversion processing. Therefore, we examined the involvement of

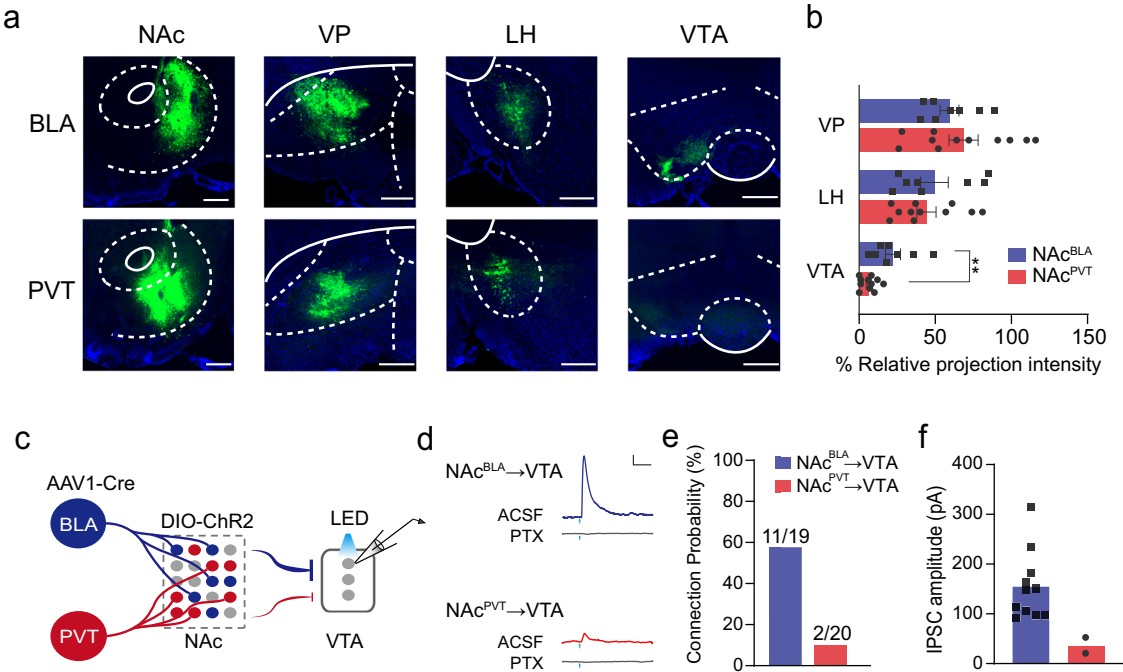

**Fig. 3 | Different projection profiles of NAc^BLA and NAc^PVT neurons.**
**a** Representative images showing projection patterns of NAc^BLA neurons (top) and NAc^PVT neurons (bottom). Scale bar: 200 μm. **b** Relative projection intensity in downstream targets for NAc^BLA ($n = 8$) and NAc^PVT ($n = 11$) neurons. VP ventral pallidum, LH lateral hypothalamus, VTA ventral tegmental area. Two-tailed Mann–Whitney test, $P = 0.0028$. Mean ± s.e.m. **c** Schematic showing the experimental design to record VTA neurons while stimulating NAc^BLA or NAc^PVT axonal terminal in VTA slice. **d** Example traces showing inhibitory postsynaptic current (IPSC) recorded from VTA neurons following brief optical stimulation of NAc^BLA (up) and NAc^PVT (bottom) terminals, with or without the presence of picrotoxin (100 mM). Scale bar: 20 pA, 100 ms. **e** Probability of synaptic connection between VTA neurons with NAc^BLA neurons ($n = 19$ from three mice) and NAc^PVT ($n = 20$ from three mice) neurons. **f** Average amplitude of IPSCs from NAc^BLA→VTA ($n = 11$) and NAc^PVT→VTA ($n = 2$) stimulation.

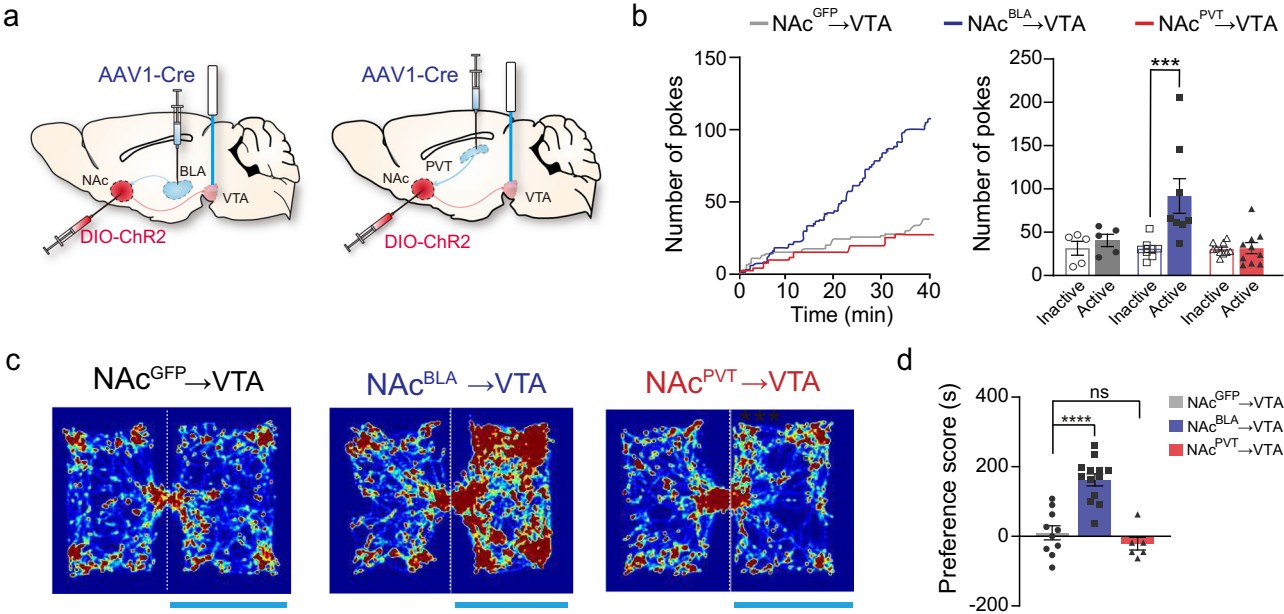

**Fig. 4 | Activation of the NAc^BLA→VTA pathway is rewarding. a** Schematic showing viral strategy to activate NAc^BLA or NAc^PVT axon terminals in the VTA. **b** Left: example cumulative curves for active nose-pokes. Right: numbers of nose-pokes to obtain optical stimulations for NAc::GFP→VTA ($n = 5$), NAc^BLA::ChR2→VTA ($n = 8$) and NAc^PVT::ChR2→VTA mice ($n = 10$). Two-way ANOVA: ChR2 x poke ($F_{(2,40)} = 5.883$, $P = 0.0058$), ChR2 ($F_{(2,40)} = 5.818$, $P = 0.0061$) followed by post hoc Sidak's test, BLA: Active vs. Inactive, $P = 0.0002$. Mean ± s.e.m. **c** Representative heatmaps for the RTPP experiments. **d** Preference scores in the RTPP test for NAc::GFP→VTA ($n = 10$), NAc^BLA::ChR2→VTA ($n = 13$) and NAc^PVT::ChR2→VTA ($n = 6$) groups. One-way ANOVA ($F_{(2,26)} = 27.51$, $P < 0.0001$) followed by post hoc Tukey's test. ****$P < 0.0001$. Mean ± s.e.m.

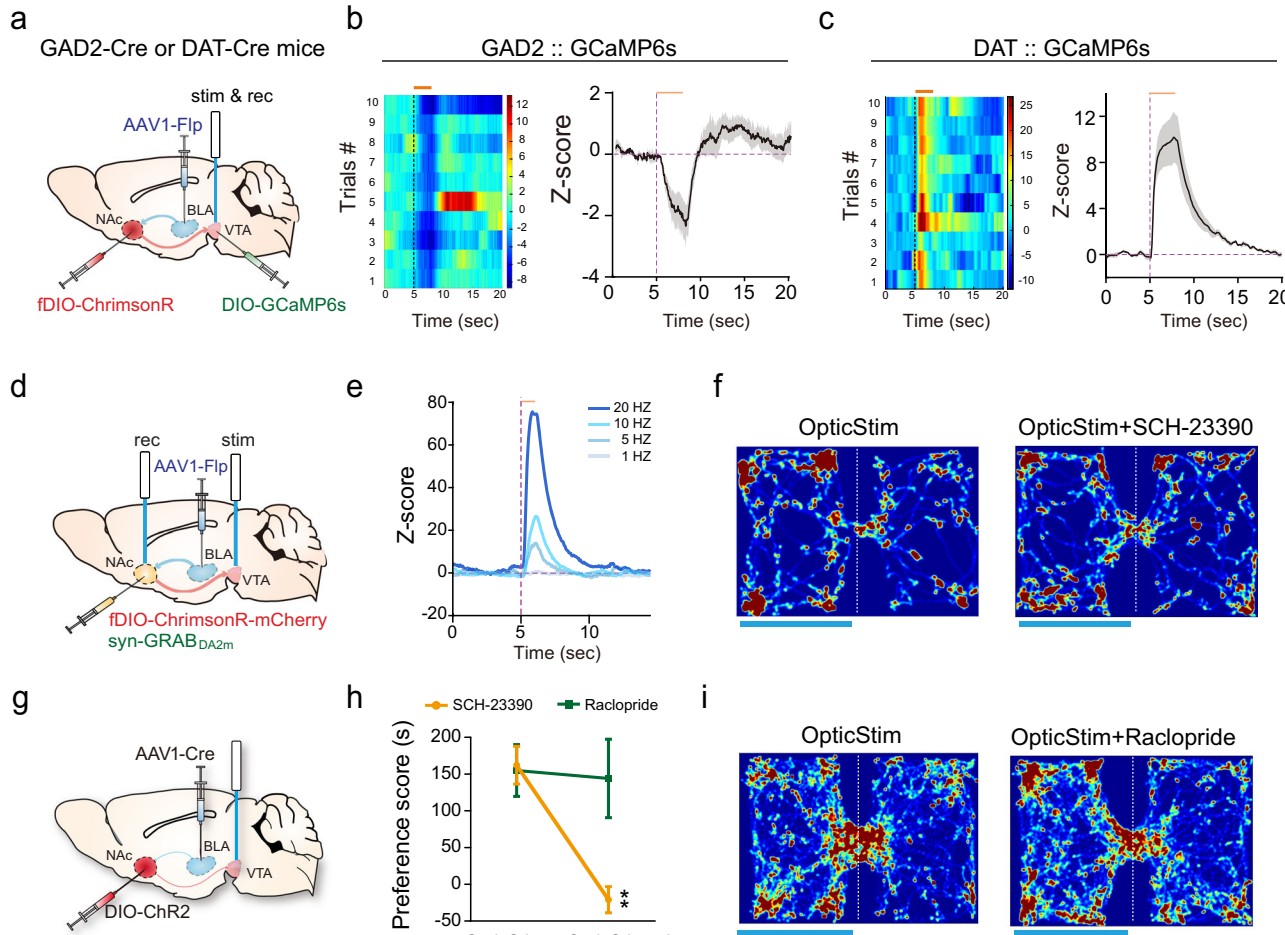

**Fig. 5 | NAc^BLA neurons mediate reward by disinhibiting dopaminergic neurons in the VTA. a** Schematic showing the experimental design to record Ca²⁺ signal from GABAergic or dopaminergic neuronal populations in VTA while delivering optogenetic stimulation to NAc^BLA→ VTA pathway. **b** Left: an example of a heatmap of calcium activities following optic stimulations in the VTA in a GAD2-Cre mouse. Right: average fiber photometry response trace of VTA GABAergic neurons elicited by optic stimulation of NAc^BLA→VTA pathway (n = 3 mice). The orange horizontal line represents light stimulation (580 nm, 20 Hz, 3 s). Shaded areas in the right panel represent ±s.e.m. **c** Left: an example of a heatmap of calcium activities following optic stimulations in the VTA in a DAT-Cre mouse. Right: Average fiber photometry response trace of VTA dopaminergic neurons elicited by optic stimulation of NAc^BLA→VTA pathway (n = 4 mice). The orange horizontal line represents light stimulation (580 nm, 20 Hz, 3 s). Shaded areas in the right panel represent ±s.e.m. **d** Schematic showing the experimental design to record

dopamine release in NAc following optogenetic stimulation of NAc^BLA→ VTA pathway. Genetically encoded dopamine sensor GRAB_DA2m was expressed in NAc^BLA neurons. Optic fiber for stimulation was placed in VTA, and optic fiber for recording was placed in NAc. (n = 6 mice). **e** Representative dopamine release in NAc recorded with GRAB_DA2m in response to 1 s optic stimulation of NAc^BLA→VTA pathway at different frequencies (580 nm, 1 s). **f** Representative heatmaps for the RTPP experiment without and with the injection of dopamine receptor 1 antagonist SCH-23390. **g** Schematic showing viral strategy to activate NAc^BLA terminals in the VTA in the RTPP test. **h** Changes in the RTPP score after administration of SCH-23390 (n = 6) or raclopride (n = 5). Two-way RM ANOVA: drug ($F_{(1,9)}$ = 6.184, P = 0.0346), drug × treatment ($F_{(1,9)}$ = 5.896, P = 0.0381), followed by post hoc Sidak's test (SCH-23390: with vs. without drug treatment, P = 0.0081). Mean ± s.e.m. **i** Representative heatmaps for the RTPP experiment without and with the injection of dopamine receptor 2 antagonist raclopride.

NAc^PVT→LH pathway in the expression of opioid withdrawal symptoms. We inhibited the NAc^PVT→LH pathway by transducing NAc^PVT neurons with the optogenetic silencing tool halorhodopsin (NpHR) and implanting optic fibers above the LH bilaterally (Fig. 6g). Optical suppression of the NAc^PVT→LH pathway significantly reduced somatic withdrawal signs (Fig. 6h), suggesting the NAc^PVT→LH pathway plays an important role in opioid withdrawal.

**NAc^PVT and NAc^BLA neurons target different cell types in the LH**
Apparently, the results described above raise the question of whether activation of the NAc^BLA→LH pathway could also elicit aversive behavior since NAc^BLA neurons also send dense projections to the LH (Fig. 3a). However, activation of the NAc^BLA→LH pathway increased the time spent in the light-paired chamber in the RTPP test, suggesting it is rewarding (Fig. 6j, k). We further confirmed this by performing an optical self-stimulation experiment and found that activation of the

NAc^BLA→LH pathway readily supports self-stimulation (Fig. 6l). In addition, we found that activation of the NAc^BLA→VP pathway does not induce significant changes in the RTPP test (Supplementary Fig. 5).

The contrasting effects of activation of the NAc^BLA→LH and the NAc^PVT→LH pathways are puzzling because both NAc^BLA and NAc^PVT populations are GABAergic neurons[1]. We hypothesized that NAc^BLA and NAc^PVT neurons might innervate distinct cell types in the LH[39]. To test this hypothesis, we performed targeted patch-clamp recordings in GAD67-GFP mice, in which GABAergic neurons were labeled with GFP (Fig. 7a). Light stimulation of NAc^PVT axon terminals in the LH evoked robust picrotoxin-sensitive IPSCs in 58.3% (7/12) of GFP-positive GABAergic neurons, while no IPSC was recorded (0/8) in GFP-negative cells (putative glutamatergic neurons) (Fig. 7b, c). In stark contrast, light stimulation of NAc^BLA axon terminals evoked robust IPSCs in 71.4% (5/7) of GFP-negative cells, while only 7.1% (1/14) of GFP-positive cells showed a small IPSC (Fig. 7d–f). Thus, our data indicate

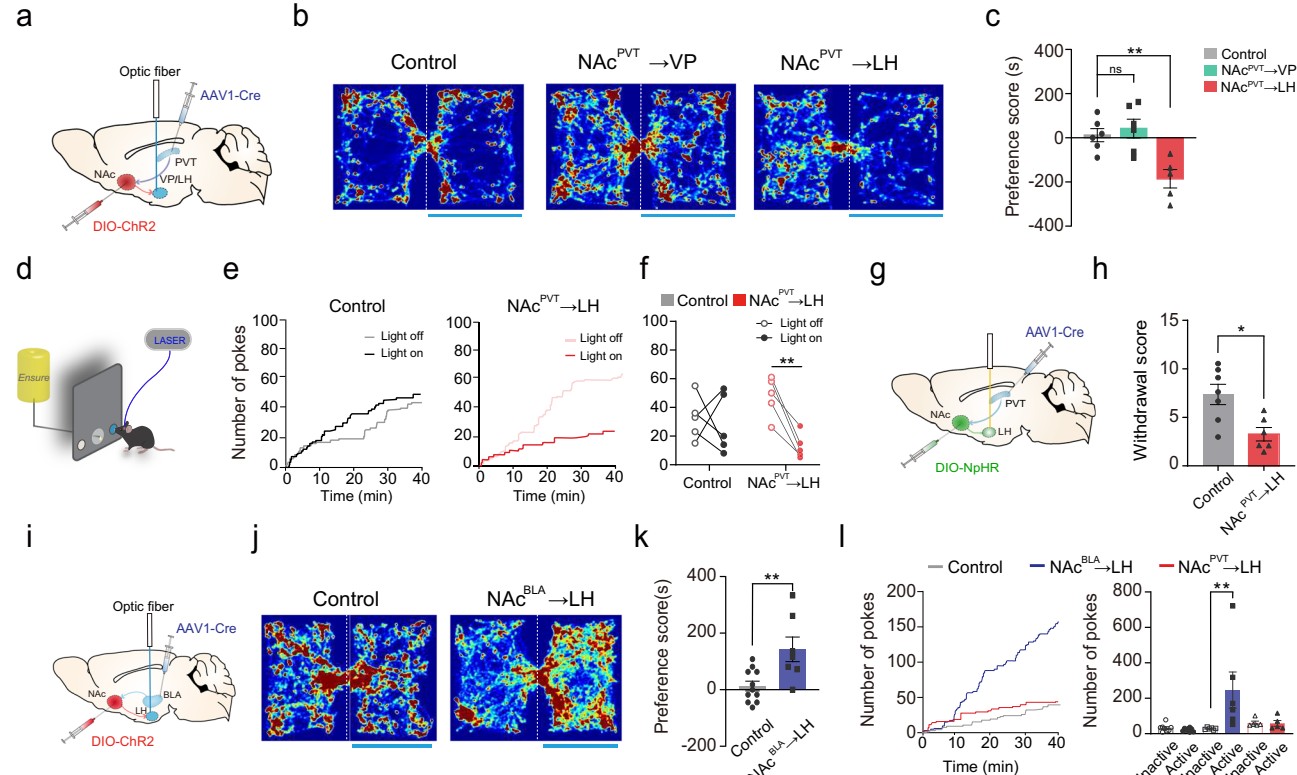

**Fig. 6 | NAc^PVT neurons mediate aversion by inhibiting GABAergic neurons in the LH. a** Schematic showing viral strategy to activate NAc^PVT axon terminals in the VP or the LH. **b** Example of heatmaps of activating NAc^PVT→VP or NAc^PVT →LH pathway in the RTPP test. **c** Preference scores for NAc::GFP ($n = 6$), NAc^PVT::ChR2→VP ($n = 6$) and NAc^PVT::ChR2→LH ($n = 5$) groups in the RTPP test. One-way ANOVA ($F_{(2,14)} = 9.842$, $P = 0.0021$) followed by post hoc Tukey's test, NAc^PVT::ChR2→LH vs. NAc::GFP ($n = 6$), $P = 0.0077$. **$P < 0.01. Mean ± s.e.m. **d** An illustration of the experimental design to test the effect of NAc^PVT →LH pathway stimulation on palatable food-seeking. Nose-pokes on either side could result in the delivery of Ensure, while only right-side pokes were coupled to laser stimulations. **e** Example cumulative curves for nose-pokes on the side assigned as the laser-coupled side. **f** Light stimulation in the LH reduced nose-pokes to earn palatable food in NAc^PVT::ChR2→LH ($n = 5$), but not in control mice ($n = 5$). Two-way RM ANOVA:

light × ChR2 ($F_{(1,16)} = 5.222$, $P = 0.0363$), followed by post hoc Sidak's test (NAc^PVT::ChR2→LH: light on vs. light off, $P = 0.0047$). **$P < 0.01. **g** Schematic showing the experimental design to bilaterally inhibit NAc^PVT terminals in the LH. **h** Withdrawal scores in naloxone-precipitated morphine withdrawal test for NAc^PVT→LH inhibition ($n = 6$) and control mice ($n = 7$). Two-tailed Mann–Whitney test, $P = 0.0221$. *$P < 0.05. Mean ± s.e.m. **i** Schematic showing viral strategy to activate NAc^BLA axon terminals in the LH. **j** Example of heatmaps for the RTPP test. **k** Rewarding effect of NAc^BLA→LH stimulation revealed by RTPP test. NAc::GFP ($n = 11$), NAc^BLA::ChR2→LH ($n = 7$). Two-tailed Mann–Whitney test, $P = 0.0052$. **$P < 0.01. Mean ± s.e.m. **l** Activation of the NAc^BLA→LH ($n = 6$), but not NAc^PVT→LH pathway ($n = 5$), supported optical self-stimulation. Two-way ANOVA: ChR2 × poke ($F_{(2,32)} = 5.130$, $P = 0.0117$) followed by post hoc Sidak's test (BLA: active vs. inactive, $P = 0.0021$. **$P < 0.01. Mean ± s.e.m.

that NAc^PVT neurons more frequently synapse on GABAergic neurons in the LH, while NAc^BLA neurons prefer to innervate glutamatergic neurons in the LH.

Since LH-projecting NAc neurons are supposed to inhibit their downstream targets, we reasoned that direct inhibition of GABAergic and glutamatergic neurons in the LH should mimic the effect of NAc^PVT→LH and NAc^BLA→LH pathway stimulation, respectively. We found that optogenetic inhibition of LH GABAergic neurons produces robust place avoidance, while inhibition of LH glutamatergic neurons promotes place preference (Fig. 7g, h). To further test whether the aversive effect of NAc^PVT activation could be attributed to the inhibition of LH GABAergic neurons, we stimulated NAc^PVT neurons while simultaneously activating LH GABAergic neurons (Fig. 7i). No behavioral aversion was observed in the RTPP test (Fig. 7j), suggesting NAc^PVT neurons mediate aversion through inhibition of LH GABAergic neurons.

## Discussion

Our results solve a long-standing puzzle of why distinct glutamatergic inputs to NAc produce opposite motivational valence and highlight the importance of input-output connectivity when dissecting NAc circuitry (Fig. 8). Specifically, NAc^BLA neurons, which receive BLA inputs, project to VTA_GABA and LH_Glu neurons to control reward-seeking behavior.

NAc^PVT neurons, which receive PVT inputs, project to LH_GABA neurons to promote aversion. These results provide an input-output connectivity framework for understanding the role of NAc subcircuits in mediating reward and aversion.

The BLA→NAc pathway has been long implicated in reward-seeking behaviors, although recent studies also suggest an important role for the BLA→NAc pathway in active avoidance[40–42]. Our results suggest that the rewarding effect of activating BLA→NAc pathways could be mediated by the VTA and the LH. We also provided evidence that BLA can regulate dopamine signals through the NAc. To our knowledge, this is the first study utilizing transsynptic viral tools to directly activate the BLA→NAc→VTA pathway and study the behavioral outcomes. The LH is reciprocally connected with the VTA, and plays an essential role in feeding, arousal, and pain regulation[39]. We found that NAc neurons could send collaterals to both downstream targets (Supplementary Fig. 4), suggesting that the BLA could coordinate activities in the LH and the VTA via the NAc to regulate positive emotions[43]. Although recent studies have also revealed transcriptional complexity and functional diversity in BLA neurons[42,44], our study focused on their connections with the medial NAc shell and analyzed the circuitry mechanism of functional divergence between NAc^BLA and NAc^PVT neurons. Further research is needed to fully resolve the input-output connectivity

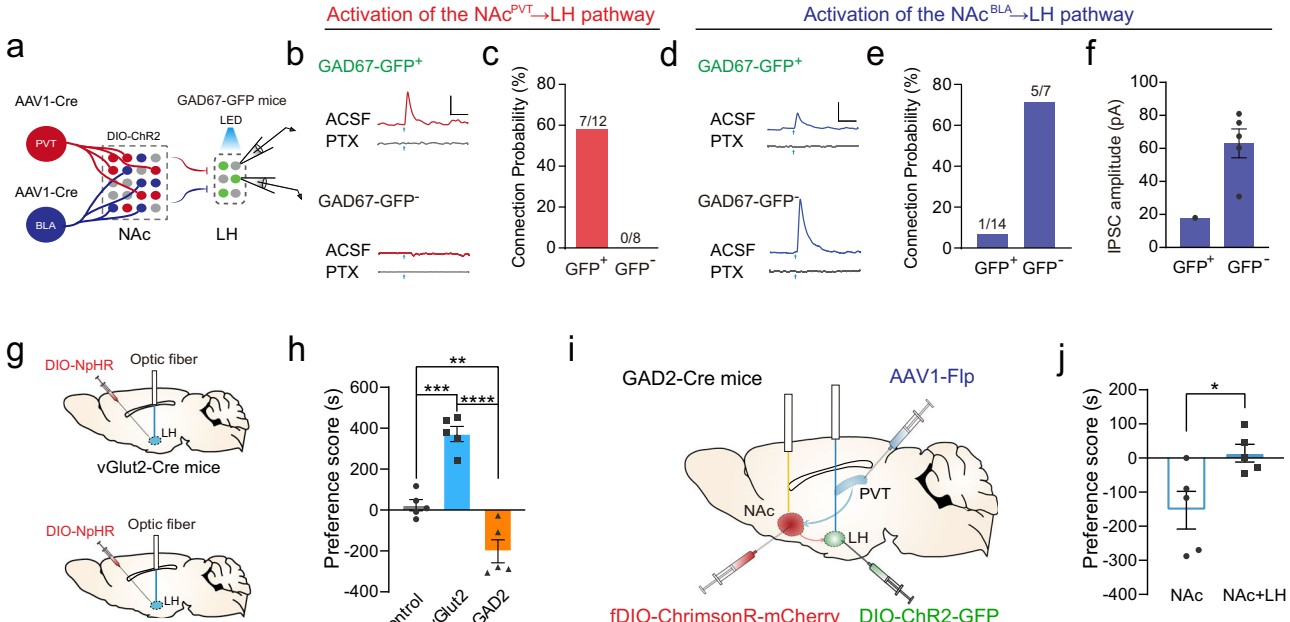

**Fig. 7 | NAc^PVT and NAc^BLA neurons innervate different types of neurons in the LH. a** Schematic showing the experimental design to record GABAergic neurons and putative glutamatergic neurons while stimulating NAc^BLA or NAc^PVT axonal terminal in LH slice from GAD67-GFP mice. **b** Example traces showing inhibitory postsynaptic current (IPSC) recorded from GFP-positive GABAergic neurons (up) and GFP-negative putative glutamatergic neurons (bottom) in the LH following brief optical stimulation of NAc^PVT terminals in the LH, with or without the presence of picrotoxin (100 mM). Scale bar: 20 pA, 100 ms. **c** Probability of synaptic connection between NAc^PVT with GFP-positive (n = 12) and GFP-negative (n = 8) neurons in the LH. **d** Example of traces showing inhibitory postsynaptic current (IPSC) recorded from GFP-positive GABAergic neurons (up) and GFP-negative putative glutamatergic neurons (bottom) in the LH following brief optical stimulation of NAc^BLA terminals in the LH, with or without the presence of picrotoxin (100 mM).

Scale bar: 20 pA, 100 ms. **e** Probability of synaptic connection between NAc^BLA with GFP-positive (n = 14) and GFP-negative (n = 7) neurons in the LH. **f** Amplitude of evoked IPSCs from GFP-positive (n = 1) and GFP-negative (n = 7) neurons following optical stimulation of NAc^BLA→LH. Mean ± s.e.m. **g** Schematic showing the experimental design for optogenetic inhibition of GABAergic and glutamatergic neurons in the LH. **h** Preference scores in RTPP test for control (n = 5), vGluT2::ChR2 (n = 5) and GAD2::ChR2 (n = 5) mice. One-way ANOVA ($F_{(2,12)} = 47.2$, $P < 0.0001$) followed by post hoc Tukey's test (vGlut2 vs. Control, $P = 0.0002$; GAD cs. Control, $P = 0.0068$). **$P < 0.01$, ***$P < 0.001$, ****$P < 0.0001$. Mean ± s.e.m. **i** Schematic showing the experimental design to simultaneously activate NAc^PVT neurons and LH GABAergic neurons. **j** Preference scores in RTPP test for mice with NAc^PVT activation alone (unfilled bar) or co-activation with LH GABAergic neurons (filled bar, n = 5). Two-tailed paired *t* test. *$P < 0.05$. Mean ± s.e.m.

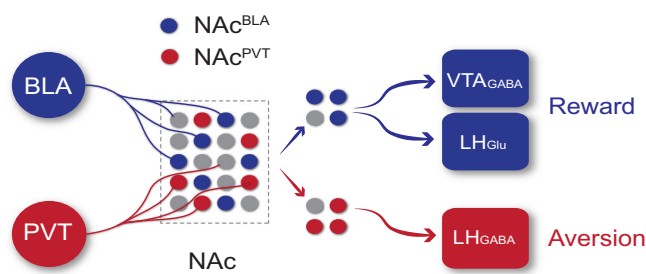

**Fig. 8 | Schematic showing roles of input-output defined parallel NAc pathways in reward and aversion processing.** The NAc^BLA neurons receive BLA inputs and project to VTA_GABA and LH_Glu neurons to control reward-seeking behavior. The NAc^PVT neurons receive PVT inputs and project to LH_GABA neurons to promote aversion.

map of different neuron populations with diverse gene expression profiles.

The evidence presented here supports previous research that the PVT→NAc pathway is highly involved in substance abuse[6,45] and aversive behaviors[46]. In this study, we further elaborated on the circuitry mechanism of drug-related behaviors mediated by NAc^PVT neurons. In accordance with the study by Engelke et al., which reported that optogenetic of the PVT-NAc suppresses reward-seeking and induces avoidance behavior, our results showed that the aversive effect induced by PVT-NAc activation could be mediated by downstream

target LH. Previously, we reported that repetitive morphine exposure enhances feed-forward inhibition onto LH-projecting NAc^PVT neurons[45]. Therefore, downregulation of the NAc^PVT → LH pathway might play a permissive role in the rewarding effect of morphine. In consistence, we showed that suppression of the NAc^PVT → LH pathway reduces the aversive effect induced by opioid withdrawal. It will be interesting to see whether neurotransmission in NAc^PVT → LH_GABA pathway is upregulated during drug abstinence or precipitated withdrawal in future investigations.

Previous investigations have shown substantial differences in drug-related behaviors between females and males[47–49]. Furthermore, sex-dependent structural and functional differences in the mesolimbic system have been revealed[50–52]. In the current study, we reported the differential roles of NAc^BLA and NAc^PVT subpopulations are independent of sex. Still, it would also be interesting for future studies to investigate the effect of gonadal hormones on NAc^BLA and NAc^PVT pathways in drug-related behaviors. In addition, using animals of both sexes would be beneficial for translating research evidence into clinical practice. Last but not least, we studied aversive behaviors mainly based on a gross summary of the time spent in different chambers in the RTPP test, but also analyzed microstructures of behaviors such as rears and self-grooming (Supplementary Fig 7), which are indicative of stress and anxiety. We believed systematic animal pose estimations would have important applications in the research of emotion and drug abuse in future studies[53].

Together, our results demonstrated input-defined, parallel NAc circuits in reward and aversion processing. Understanding the circuitry

mechanism that orchestrates opposing motivations will guide the future development of treatment strategies for addiction and other neuropsychiatric disorders.

# Methods

## Animals

Mice aged 5–12 weeks were used in the experiments. Mice were housed at 22–25 °C under a 12-h light–dark cycle. All husbandry and experimental procedures in this study were approved by the Animal Care and Use Committees at the Shenzhen Institute of Advanced Technology (SIAT), Chinese Academy of Sciences (CAS). C57BL/6j mice were purchased from Charles River Laboratories in Beijing and Hangzhou.GAD2-Cre (JAX Stock No: 010802), DAT-Cre (JAX Stock No: 006660), vGlut2-Cre (JAX Stock No:016963), Ai14 (JAX Stock No: 007908), and R26R-EYFP (JAX Stock No: 006148) were used in the current study. GAD67-GFP mice were originally from Dr. Nobuaki Tamamaki's lab, and FSF-tdTomato (derived from Ai65 by breeding with CMV-Cre to remove the LoxP-STOP-loxP cassette) mice were originally from Dr. Z. Josh Huang's lab. Cre/Flp double-reporter mice were obtained by crossing R26R-EYFP mice with FSF-tdTomato mice. Mainly male mice were used in the current study if not indicated otherwise.

## Virus and reagents

Virus used in the current study were purchased from Taitool company: AAV1-syn-Cre (S0278); AAV1-syn-Flp (S0271); AAV2/9-hEF1a-DIO-hChR2(H134R)-mCherry (S0170); AAV2/9-hEF1a-DIO-hChR2(H134R)-EYFP (S0199); AAV2/8-hEF1a-DIO-EYFP (S0196); AAV2/9-hEF1a-DIO-eNPHR3.0-mCherry (S0197); AAV2/9-CAG-DIO-EGFP-2A-TeNT (S0235); AAV2/9-hEF1a-fDIO-ChrimsonR-mCherry (S0384); AAV2/9-hsyn-FLEX-Gcamp6s (S0226); AAV2/9-hEF1a-DIO-eNPHR3.0-mCherry (S0178); AAV2/9-hEF1a-DIO-eNPHR3.0-EYFP (S0852). The titer of AAV1-syn-Cre virus is above $1 \times 10^{13}$ GC/mL, while the titer of other viruses ranges from 2 to $5 \times 10^{13}$ GC/mL. We used CTB-488 (Invitrogen™,C34775), CTB-555 (Invitrogen™,C34776) and CTB-647 (Invitrogen™,C34778) at 1 mg/mL. Morphine was purchased from China National Accord Medicines and naloxone was purchased from Sigma-Aldrich.

## Stereotaxic surgeries

Mice were anesthetized with i.p. injections of pentobarbital (80 mg/kg) and positioned in a stereotaxic frame (RWD, 68019). Standard procedures were performed as previously described[45]. Virus injections were under the control of LEGATO syringe pump (KD scientific, 788130). Hamilton syringe (#65460-05,10 μL) was used for injections. The volume of AAV1-Cre injections in the BLA or the PVT was 200 nL, while the volume of viral injections in the NAc was 300–400 nL per side. The CTB injections in the VP, LH or the VTA were 200 nL. The injection rate was 50–70 nL/min. After the injection was done, we waited for at least 5 min to avoid backflow and then pull out the syringe slowly. For anatomical tracing experiments, the incision was stitched after the injection by using a surgical suture. For stimulation experiments, we did only one injection in the PVT or the BLA. For suppression experiments, injections were performed on both sides of the mouse brains. The following coordinates were used for virus injection: NAc (AP + 1.5 mm; ML +/− 0.67 mm; DV −4.6 mm from the bregma), BLA (AP −1.5 mm; ML +/− 3.25 mm; DV −4.85 mm from the bregma), PVT (AP −1.4 mm; ML −0.3 mm; DV −3.2 mm from the bregma; 5 ~ 6° toward the midline), LH (AP −1.4 mm; ML +/− 1.4 mm; DV −5.1 mm from the bregma), VP (AP + 0.7 mm; ML +/− 1.3 mm; DV −4.8 mm from the bregma), VTA (AP −3.4 mm; ML −0.4 mm; DV −4.5 mm from the bregma). For optogenetic experiments, optic ferrules (O. D. 1.25 mm, Fiber Core 200 μm, NA 0.37) were implanted at 400–500 μm above the injection sides. For fiber photometry, optic fibers were placed 200 μm above the targeted region. Optic fibers were fixed to the skull

with light-cured dental resin. Animals were allowed to recover for at least 3 weeks before experiments.

## Histology and in situ hybridization

Mice were euthanized with an overdose of pentobarbital sodium and transcardially perfused with phosphate-buffered saline (PBS, pH 7.4) followed by 4% paraformaldehyde (PFA) in PBS. Brain tissues were dissected and postfixed for 1–2 h in 4% PFA in PBS at room temperature. After being dehydrated for 24–48 h in 30% sucrose, brain tissues were embedded in the Tissue-Tek OCT compound (Sakura) on dry ice before sectioning. Brains were cut into 50-μm sections with a cryostat (Leica). Free-floating cryosections were collected in PBS. Brain sections were first washed in PBS (3 × 10 min), then blocked at room temperature with 5% normal bovine serum in PBST (0.3% Triton X-100) and then incubated with the primary antibody (1:1000, Anti-GFP, Thermofisher A11122; 1:1000, Anti-Cre, Millipore, MAB2130) overnight at 4 °C. Brain sections were washed in PBST (3 × 10 min), followed by incubation for 2 h with fluorophore-conjugated secondary antibody (1:1000, Goat anti-Rabbit-488 #111-547-003 or Donkey anti-Mouse-Cy3,#715-165-150, Jackson Immuno) and finally counterstained with DAPI (1:30,000).

For in situ RNA hybridization, we used RNAscope multiplex fluorescent reagent kit v2 assay, including sample preparation and pretreatment. Briefly, mouse brains were cut into 14–20 μm sections with a cryostat (Leica) and mounted onto SuperFrost Plus microscope slides. The brain sections were dried at 39 °C for 2 h, rinsed in 1× PBS, treated with 3% hydrogen peroxide in methanol for 5 min, treated with TR buffer for 15 min at 100 °C, dehydrated with ethanol and then treated with RNAScope protease III for 15 min at 40 °C. The rest of the staining procedures were performed following the manufacturer's protocols for fixed-frozen tissue samples and I HybEZ™ oven (Advanced Cell Diagnostics, Inc). In the current study, we used probes Drd1 (Advanced Cell Diagnostics, #13488) and Drd2 (Advanced Cell Diagnostics, #13489).

## Imaging, cell counting, and cell distribution analysis

To visualize the downstream projection brain regions of NAc neurons receiving BLA and PVT input, 50-μm brain sections were photographed with Olympus Virtual Slide Microscope (VS120-S6-W) using ×10 objective. We found that the major downstream projection brain regions are the VP, LH, and VTA. Three representative images were selected for each downstream brain region for every mouse, and the fluorescence intensity was calculated by ImageJ software. To quantify the co-localization of virus-expressing neurons with endogenous mRNA (D1/ D2) or CTB-647/CTB-488, three or four representative ×20 images of each mouse were selected. Co-localization was identified by eye.

## Electrophysiological recordings

The procedures for preparing acute brain slices and performing whole-cell recordings with optogenetic stimulation were similar to the previous study[6]. Briefly, animals were transcardially perfused with ice-cold choline-based solution containing (in mM) 110 choline chloride, 2.5 KCl, 0.5 CaCl₂, 7 MgCl₂, 1.3 NaH₂PO₄, 1.3 Na-ascorbate, 0.6 Na-pyruvate, 25 glucose and 25 NaHCO₃, saturated with 95% $O_2$ and 5% $CO_2$, under isoflurane anesthesia. Coronal 250–300 μm slices containing the LH or VTA were prepared using a vibratome (VT-1000S, Leica), and were incubated in 32 °C oxygenated artificial cerebrospinal fluid (in mM: 125 NaCl, 2.5 KCl, 2 CaCl₂, 1.3 MgCl₂, 1.3NaH₂PO₄, 1.3 Na-ascorbate, 0.6 Na-pyruvate, 25 glucose, and 25 NaHCO₃) for at least 30 min before recording. Patch pipettes (2–5 MΩ) were filled with a Cs-based low Cl⁻ internal solution containing (in mM) 135 CsMeSO₃, 10 HEPES, 1 EGTA, 3.3 QX-314, 4 Mg-ATP, 0.3 Na-GTP, 8 Na₂-phosphocreatine, 290 mOsm kg⁻¹, adjusted to pH 7.3 with CsOH. The whole-cell voltage-

clamp recording was performed at room temperature with a Multiclamp 700B amplifier and a Digidata 1440 A (Molecular Devices). Data were sampled at 10 kHz and analyzed with Clampfit (Molecular Devices) or MATLAB (MathWorks). A blue light-emitting diode (470 nm, Thorlabs) controlled by digital commands from the Digidata 1440 A was used to deliver photostimulation. To record light-evoked IPSCs, a blue light pulse (2 ms, 0.5–2 mW) was delivered through an optic fiber to illuminate the entire field of view. The IPSCs were recorded at a holding potential of 0 mV. The EPSCs were recorded at a holding potential of −70 mV. To block IPSCs, picrotoxin (PTX 100 μm) was added into the recording chamber through the perfusion system and incubated for at least 5 min.

### Fiber photometry

To perform fiber photometry recording and optogenetic stimulation in the same brain region, we used the device from THINKERTECH (QAXK-FPS-SS-LED-OG). A 580 nm stimulation laser (10 mW) coupled to a 470 nm blue LED light (30 μW) was delivered through a single implanted ferrule. Analysis of the signal was performed with custom-written MATLAB codes (Supplementary Code 1). The Z-score was calculated as $(x - \mu)/\sigma$, using the mean and standard deviation of the signal 5 s before the stimulation.

### Behavioral assays

Before behavioral examinations, animals were acclimated to the experimental room for at least 30 min. For stimulation experiments, animals were tested in both the RTPP test and the self-stimulation test, but on different days. For suppression experiments, animals were used in only one behavioral assay.

**Real-time place preference (RTPP).** In the RTPP test, a custom-made two-chamber apparatus was used. Before the experiments, the animals were gently attached to the optical fiber. On the first day, mice were allowed to explore the apparatus for 15 min without being stimulated by light. On the second day, we designated the counterbalanced side as the stimulation side. When the mouse entered the stimulation side, 20 Hz laser stimulation (473 nm, 20-ms pulse duration, 5–10 mW mm$^{-2}$ per side) was delivered. As long as the mouse returned to the other side, the laser stimulation was turned off. The place preference score was calculated as the time spent on the simulation side during the test, by subtracting that from the baseline. We used female mice to repeat the experiments in Fig. 2f (data shown in Supplementary Fig. 6).

**Optical intracranial self-stimulation.** The optical self-stimulation test was performed in the operant chamber (Anilab). Mice were allowed to explore the chamber with optic fiber attached for 45 min. When the mouse nose-poked the left-side hole on the wall, it triggered a train of light stimulation (20 Hz, 20 ms pulse duration) that lasted for 1.5 s. Nose-pokes on the right side did not lead to laser stimulation. We used female mice to repeat the experiments in Fig. 2c (data shown in Supplementary Fig. 6).

**Palatable food-seeking.** Mice had unlimited access to standard chow and water. For the TeNT silencing experiments in Fig. 2, the mice were first habituated to the operant behavioral chamber (Anilab) for three daily sessions (30 min). A drop (10–12 μl) of Ensure was delivered from the lickometer spout in the behavioral chamber every 10 s in the habituation sessions. Mice that readily learned to lick the Ensure drops were subjected to the FR = 1 training for 3 days, during which a right-side nose-poke would trigger the delivery of a drop of Ensure. The pump would be turned off for 2 s once the delivery of Ensure was accomplished. For the optogenetic stimulation experiments in Fig. 2, mice were first trained on the FR = 1 schedule without stimulation, during which either a left-side or a right-side nose-poke could be given a drop of Ensure reward. During the stimulation session, one side of

the nose-poke would also trigger a 1.5-s laser stimulation. The side of stimulation was counterbalanced.

**Naloxone-precipitated morphine withdrawal.** Mice received daily i.p. injection of morphine with escalating doses at 10, 20, 30, 40, 50, and 50 mg/kg in their homecages. Two hours after the last morphine injection on day 6, mice were injected with naloxone (5 mg/kg, i.p.) and placed in an open chamber in sound-proof boxes. Withdrawal symptoms were recorded for 20 min. Physical signs were manually counted by students blinded from the experimental design. To make each sign of equal weight, a relative withdrawal score for each physical sign was normalized by the maximum number from all the animals. That is, each physical sign was given a possible score from 0–1 for an individual mouse. And the global withdrawal score was calculated as diarrhea *3 + rearing*3 + tremor*5 + grooming*2 + jump*7[54,55].

### Statistics and reproducibility

Statistical analyses were performed using Prism 8.0 (GraphPad Software). Experiments shown in Fig. 3d, j, Supplementary Figs. 1a, b and 2 were repeated independently in at least five animals with similar results, which were also summarized in Supplementary Fig. 8.

### Reporting summary

Further information on research design is available in the Nature Research Reporting Summary linked to this article.

## Data availability

All raw data supporting the findings of this study are available from the corresponding author upon request. Source data are provided with this paper.

## Code availability

Custom scripts for fiber photometry experiments are included in the supplementary materials of the online version.

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

## Acknowledgements

We thank Dr. Erwin Neher, Dr. Jianyuan Sun, Dr. Guoqiang Bi and Dr. Xiaoke Chen for helpful discussions, and Dr. Yulong Li for providing us with the DA$_{2m}$ sensor. We thank Dr. Wenyu Qian for help with tracing

experiments. This work was supported by Science and Technology Innovation 2030 - Major Project (2021ZD0203500), National Natural Science Foundation of China (81922024, 82171492, 31900735, 31970971, and 32171087), Science, Technology and Innovation Commission of Shenzhen Municipality (RCJC20200714114556103 & ZDSYS20190902093601675), Guangdong Provincial Key Laboratory of Brain Connectome and Behavior (2017B030301017).

## Author contributions

Y.Z. conceived the study. K.Z., H.X., S.L., and Y.Z. designed the experiments and analyzed data. K.Z., H.X., and S.L. conducted tracing and behavior experiments with the help of M.H. K.Z. conducted fiber photometry recording experiments with the help of X.D. S.J and G.H conducted patch-clamp recording experiments. K.Z. and Y.Z. wrote the manuscript. All authors reviewed the manuscript.

## Competing interests

The authors declare no competing interests.
