## [Peer Review File · Nature Communications]

Title: Reward and aversion processing by input-defined parallel nucleus accumbens circuitsREVIEWER COMMENTS

Reviewer #1 (Remarks to the Author):

The authors report findings from an impressive series of experiments examining how different populations of nucleus accumbens (NAc) neurons that receive input from either the basolateral amygdala (BLA) or PV thalamus regulate appetitive vs aversively motivated behaviors, and how activation of these distinct cell groups exert their effect via different projection pathways. In general, NAc neurons receiving BLA inputs mediated reward/appetitive motivation (assessed with self stimulation, real time place preference and food-seeking procedures). This in turn was driven by outputs of these cells to the VTA (which inhibited GABA cells there and disinhibited Dopamine cells, increasing dopamine release), and by acting on non-GABAergic cells in the lateral hypothalamus. In contrast, NAc neurons receiving PV thalamus inputs appeared to mediate aversion mediated aversion (indexed with the same assays as above as well as morphine withdrawal) and seemed to do so via actions on GABA cells in the hypothalamus. The authors also provide additional novel and sure to be useful data on some of the projection targets of these different cell populations in the NAc, their dopamine receptor expression and their segregation.

These are a very interesting set of findings that will further enhance our understanding of how different NAc circuits regulate behaviors of opposing motivational valence. Some of the findings (eg, BLA-NAc inputs mediating reward, PV THal inputs mediating aversion) have been reported previously but the additional data characterizing the output pathways are novel and intriguing. There are a few issues that the authors will need to address, particularly with respect to integrating their findings with the existing literature.

1) When discussing the projection profiles of BLA-NAc vs PV Thal-NAc cells (lines 147-165) I was a bit confused with the discussion of the “overlap” between the projections. For example, it is stated that “found that LH projecting NAc neurons showed similar portions of overlap with NAc BLA and NAc PVT neurons.” Yet, earlier in the manuscript it was noted that there is very little overlap with in terms of NAc cells receiving either BLA vs PV thal inputs. So the confusion is how can there be that much overlap with the outputs of these cells if one is discussing what appears to be mostly separate population of neurons? The wording here needs to be clarified so its clear what they mean by overlap here and how this corresponds to the lack of overlap in terns of NAc neurons that receive thalamic or BLA input.

2) In the extended data set, when showing the effects of activation of BLA inputs to the NAc, they only show IPSPs. Were EPSPs observed as well (as has been reported in numerous studies?) This should be noted.

3) Also in the extended data set, Fig 5 showing the effects of stimulation of BLA-NAc outputs to the VP does not appear to be referred to at all in the text. This should be mentioned. In this regard, statistics should be reported and perhaps the individual data points plotted here as the effect is listed as not significant but there certainly appears to be an effect here.

4) There were a LOT of optogenetic stimulation/suppression experiments in a variety of brain nuclei and subnuclei, but no histology figures are reported showing the specific location of fiber placements. Were these confirmed? This should be added to the supplemental sections so it is clear where in these nuclei these effects were evoked. To that end, were there any missed placements? This should be mentioned as well.

5) Although the effects reported here on BLA-NAc circuits show it mediates reward behavior, there are other data showing this pathway also can mediate active avoidance (eg, Ramirez F, Moscarello JM, LeDoux JE, Sears RM. J Neurosci. 2015). Thus, the impression left in their discussion is that this circuit is just about reward, and it does not integrate these findings with the literature on avoidance. I think the discussion should be tempered a bit, acknowledging its role in avoidance as well as reward and not making as much broad statements of the valence-specific involvement of this pathway in these behaviors.

6) In a similar vein, I was very surprised that the work of Engelke and Do-Monte (Nature Commun, 2021) were not mentioned at all in this manuscript, as this group also revealed a role for the PV thalamus pathway in aversive behaviors. This needs to be integrated in the discussion FH.

Reviewer #2 (Remarks to the Author):

The manuscript applies sophisticated approaches to investigate whether BLA and PVT inputs to the NAc differentially control reward and aversion. The approaches are state-of-the-art and involve anatomical, electrophysiological, and behavioral methods. The methods rely heavily on AAV-mediated transsynaptic tagging for many of the experiments supporting that the BLA and PVT largely synapse on a separate population of NAc neurons with PVT connections driving aversion whereas BLA projections driving reward via the LH and VTA, respectively. The approaches are sophisticated and the results interesting. However, the lack of details in the results as presented make this reviewer not as confident as one should be to recommend publication of this paper in this prestigious journal. The paper is largely anatomical connectivity based but the data and methods as presented fall short.

Based on the methods, it appears that only one injection is done in the PVT and BLA. It is important to show what specific areas were transduced as this will have impact on the location of NAc cells transsynaptically transduced. The pattern of these cells does not conform to what is known about projections being studied.

Most of the anatomical experiments lack credibility because of the lack of details provided in the methods and the display of the results. The anatomical mappings are of poor quality and do not show the details necessary to judge as a reviewer or potential reader. For example, Fig. 1 needs to show high magnification images of the cells labeled by the PVT and BLA is necessary. Images are often super saturated and misleading and the patterns of labeling shown in the extended data do not appear to

follow the boundaries associated the PVT anterograde tracing. Anatomical delineations are not adequate and do not instill confidence. For example, regional cell clusters would be expected in the NAc following targeting of the PVT and BLA.

More details on the location of the optical fibers in the NAc are necessary considering that the transduced cells throughout the extent of the NAc appears to be occurring. The PVT and BLA fibers in the NAc are likely part of a highly collateralized system to project to other areas. Optostimulation of these fibers in the NAc is likely to result in stimulation collaterals to many areas other than the NAc. Control experiments with fibers placed in other regions receiving projections from the PVT and BLA is required for experiments to be conclusive.

Reviewer #3 (Remarks to the Author):

This interesting study from Zhou et al uses an anterograde tracing and optogenetics approach to study the roles of distinct NAc neuron subpopulations in behavior. They find that NAc neurons receiving inputs from BLA are important for positive reinforcement while those receiving inputs from PVT mediate aversive behaviors. The circuit tracing is well-performed and included a nice experiment using Cre/Flp double reporter mice to show that NAcBLA and NAcPVT neurons were (mostly) distinct neuronal populations. The optogenetic experiments were well-controlled and showed that the NAcBLA and NAcPVT subpopulations played different roles in behavior. The authors then built further upon the circuit tracing to show that NAcBLA and NAcPVT subpopulations had different projection profiles and were thus part of two distinct circuits. This is a nice study, and I have only a few important suggestions for edits.

1) Only male mice were used in this study. For many reasons, this is not ideal. Going forward, I strongly urge the authors to include female mice in all their experiments. However, I realize some of these experiments involved complex breeding strategies and it would be very time consuming to repeat those experiments for this manuscript. Therefore, I strongly recommend that the authors be transparent throughout the text that their experiments were performed only in males. This should be made clear in the abstract, introduction, and results. The authors should also add a paragraph to the discussion explaining why it is critical for these experiments to be repeated in females.

2) The discussion is extremely short and mainly rehashes the results. This section needs to be improved. In addition to discussing the need females, the authors should discuss other future directions. One possibility is that they consider whether their behavioral outcomes match the precision of their neural circuit approaches. The approaches used to define and modulate the specific neuronal subpopulations in NAc are state of the art. However, I would argue that the behavioral outcomes used (such as RTTP) rely on less precise summary statistics. Perhaps future experiments should study the microstructure of behavior during behavior. Pose estimation could be useful to the authors considering the precision of their circuit approaches.

Reviewer #1:

The authors report findings from an impressive series of experiments examining how different populations of nucleus accumbens (NAc) neurons that receive input from either the basolateral amygdala (BLA) or PV thalamus regulate appetitive vs aversively motivated behaviors, and how activation of these distinct cell groups exert their effect via different projection pathways. In general, NAc neurons receiving BLA inputs mediated reward/appetitive motivation (assessed with self stimulation, real time place preference and food-seeking procedures). This in turn was driven by outputs of these cells to the VTA (which inhibited GABA cells there and disinhibited Dopamine cells, increasing dopamine release), and by acting on non-GABAergic cells in the lateral hypothalamus. In contrast, NAc neurons receiving PV thalamus inputs appeared to mediate aversion mediated aversion (indexed with the same assays as above as well as morphine withdrawal) and seemed to do so via actions on GABA cells in the hypothalamus. The authors also provide additional novel and sure to be useful data on some of the projection targets of these different cell populations in the NAc, their dopamine receptor expression and their segregation.

These are a very interesting set of findings that will further enhance our understanding of how different NAc circuits regulate behaviors of opposing motivational valence. Some of the findings (eg, BLA-NAc inputs mediating reward, PV THal inputs mediating aversion) have been reported previously but the additional data characterizing the output pathways are novel and intriguing. There are a few issues that the authors will need to address, particularly with respect to integrating their findings with the existing literature.

We appreciate the reviewer's recognition of our work.

1) When discussing the projection profiles of BLA-NAc vs PV Thal-NAc cells (lines 147-165) I was a bit confused with the discussion of the "overlap" between the projections. For example, it is stated that "found that LH projecting NAc neurons showed similar portions of overlap with NAc BLA and NAc PVT neurons." Yet, earlier in the manuscript it was noted that there is very little overlap with in terms of NAc cells receiving either BLA vs PV thal inputs. So the confusion is how can there be that much overlap with the outputs of these cells if one is discussing what appears to be mostly separate population of neurons? The wording here needs to be clarified so it is clear what they mean by overlap here and how this corresponds to the lack of overlap in terms of NAc neurons that receive thalamic or BLA input.

We apologized for the confusion when discussing the "overlap" between the projections in lines 147-165. There is indeed very little overlap between NAc neurons receiving BLA inputs (NAc^{BLA}) and receiving PVT inputs (NAc^{PVT}), as reported earlier in the manuscript. The "overlap" here refers to the overlap between NAc^{BLA} / NAc^{PVT} with LH-projecting NAc neurons. What we observed here is that, first, NAc^{BLA} neurons project to both LH and VTA, while NAc^{PVT} neurons project to LH but not VTA. Second, the overlap ratio between NAc^{BLA} neurons and LH-projecting NAc neurons is similar to the overlap between NAc^{PVT} neurons and LH-projecting NAc neurons, while the overlap ratio between NAc^{PVT} neurons and VTA-projecting NAc neurons is much less than the overlap between NAc^{BLA} neurons and VTA-projecting NAc neurons. Third, this connection specificity was further confirmed by targeted slice electrophysiology combined with optogenetic stimulation. We revised this paragraph substantially to clarify our claim in the text (*Line: 160-174*). All the line numbers in the rebuttal letter refer to the revised version of the manuscript with tracked changes.

2) In the Supplementary set, when showing the effects of activation of BLA inputs to the NAc, they only show IPSPs. Were EPSPs observed as well (as has been reported in numerous studies?) This should be noted.

In **Supplementary Fig. 1f-h**, we expressed ChR2 in NAc^{BLA} neurons and performed

whole-cell patch clamp from neighboring NAc neurons to verify the efficacy of TeNT-mediated synaptic silencing. Since the vast majority of NAc neurons were GABAergic medium spiny neurons, we didn't record EPSCs in previous experiments. In a new set of experiments, we recorded EPSCs at the holding potential of -70 mV. As expected, optogenetic stimulation of NAc^{BLA} neurons did not evoke reliable EPSCs from all recorded neighboring neurons (n = 7). We added this piece of data into **Supplementary Fig. 1i-k**.

Rebuttal Fig.1 Optical stimulation of NAc^{BLA} neurons evoked IPSC but not EPSC in neighboring neurons.

a, Schematic showing the experimental design to record postsynaptic currents in neighboring neurons following optical stimulation of NAc^{BLA} neurons.

b, Example traces of light-evoked postsynaptic currents.

c, Amplitudes of light-evoked IPSCs and EPSCs (n = 7 cells). Paired t-test, **P < 0.01. Mean ± s.e.m.

In Supplementary Figure 1c-e, we expressed ChR2 in BLA/PVT neurons, and performed whole-cell patch clamp recording from NAc^{BLA} or NAc^{PVT} neurons, to verify the functional synaptic connectivity. In the revised manuscript, we reported both EPSCs and IPSCs. The latencies of IPSCs (NAc^{BLA}: 8.90 ± 0.46 ms, NAc^{PVT}: 8.81 ± 0.88; mean ± s.e.m) were much longer than that of EPSCs (NAc^{BLA}: 3.65 ± 0.21 ms, NAc^{PVT}: 3.56 ± 0.10; mean ± s.e.m), suggesting a disynaptic feedforward inhibition (Rebuttal Fig.2d). Application of CNQX blocked both the EPSC and IPSC (**Rebuttal Fig. 2b**), confirming the disynaptic origin of the IPSC. Application of TTX and 4-AP solely blocked the IPSC while leaving the EPSC intact (**Rebuttal Fig. 2c**), suggesting the monosynaptic origin of EPSC and disynaptic origin of IPSC. This piece of data has been added to **Supplementary Fig. 1c-e** in the revised manuscript.

Rebuttal Fig.2 Optical stimulation of BLA or PVT terminals evoked monosynaptic EPSC in NAc^{BLA} and NAc^{PVT} neurons, respectively.

- a, Schematic showing the experimental design to verify functional synaptic connectivity between BLA/PVT and anterograde labeled neurons in NAc.
- b, Example trace of light-evoked postsynaptic currents with and without CNQX.
- c, Example trace of light-evoked postsynaptic currents with and without TTX & 4-AP.
- d, Latencies of light-evoked IPSCs and EPSCs (BLA: n = 8 cells, PVT: n = 7 cells). Two-way ANOVA ****P < 0.0001. Mean ± s.e.m.
- e, Amplitudes of light-evoked IPSCs and EPSCs (BLA: n = 8 cells, PVT: n = 7 cells). Mean ± s.e.m.

3) Also in the Supplementary set, Fig 5 showing the effects of stimulation of BLA-NAc outputs to the VP does not appear to be referred to at all in the text. This should be mentioned. In this regard, statistics should be reported and perhaps the individual data points plotted here as the effect is listed as not significant but there certainly appears to be an effect here.

Following the reviewer's request, we presented the individual data points and reported the statistics for **Supplementary Fig. 5 (Rebuttal Fig. 3)**. This data was mentioned in the revised manuscript (Line: 230-231).

Rebuttal Fig.3 Optical stimulation of NAc^{BLA}-VP pathway in the RTPP test.

- a, Schematic showing the experimental design to activate NAc^{BLA} terminals in the VP in the RTPP test.
- b, Preference scores in the RTPP test for mice with NAc^{BLA}→VP pathway activation (n = 6 for each group). Mann-Whitney test. ns, not significant.

4) There were a LOT of optogenetic stimulation/suppression experiments in a variety of brain nuclei and subnuclei, but no histology figures are reported showing the specific location of fiber placements. Were these confirmed? This should be added to the supplemental sections so it is clear where in these nuclei these effects were evoked. To that end, were there any missed placements? This should be mentioned as well.

We thank the reviewer for raising this important point. We illustrated the injection sites as well as the locations of fiber placement in **Supplementary Fig. 8 (Rebuttal Fig. 4)**. As indicated in the Reporting Summary file, we excluded about 5% animals in which the virus expression or fiber placement was not present in expected brain regions.

Rebuttal Fig.4 Virus expression and optic fiber locations for stimulation/suppression experiments.

a-b, Schematics showing the viral expression in the BLA and the PVT from anterior to posterior locations, respectively. Schematics in the second row represent the injection sites in reporter lines, corresponding to experiments in Fig 1 and Supplementary Fig 2-3. Schematics in the third row represent the injection sites in wild type animals in which immunostaining for Cre was performed, related to the experiments in Fig 2c,2f, 4b, and 4d.

c, Schematics showing viral expression for stimulation/suppression experiments in the NAc from anterior to posterior locations.

d-g, Schematics showing optic fiber locations in the NAc, VTA, LH, and VP for stimulation/suppression experiments.

5) Although the effects reported here on BLA-NAc circuits show it mediates reward behavior, there are other data showing this pathway also can mediate active avoidance (eg, Ramirez F, Moscarello JM, LeDoux JE, Sears RM. J Neurosci. 2015). Thus, the impression left in their discussion is that this circuit is just about reward, and it does not integrate these findings with the literature on avoidance. I think the discussion should be tempered a bit, acknowledging its role in avoidance as well as reward and not making as much broad statements of the valence-specific involvement of this pathway in these behaviors.

Following the reviewer's suggestion, we discussed and acknowledged the role of BLA→NAC pathway in active avoidance (*Line: 266 - 268*).

6) In a similar vein, I was very surprised that the work of Engelke and Do-Monte (Nature Commun, 2021) were not mentioned at all in this manuscript, as this group also revealed a role for the PV thalamus pathway in aversive behaviors. This needs to be integrated in the discussion FH.

Following the reviewer's suggestion, we discussed the work of Engelke and Do-Monte (Engelke et al., 2021) to acknowledge the role of PVT→NAC pathway in aversive behaviors (*Line: 284, 286 - 289*).

Reviewer #2:

The manuscript applies sophisticated approaches to investigate whether BLA and PVT inputs to the NAc differentially control reward and aversion. The approaches are state-of-the-art and involve anatomical, electrophysiological, and behavioral methods. The methods rely heavily on AAV-mediated transsynaptic tagging for many of the experiments supporting that the BLA and PVT largely synapse on a separate population of NAc neurons with PVT connections driving aversion whereas BLA projections driving reward via the LH and VTA, respectively. The approaches are sophisticated and the results interesting. However, the lack of details in the results as presented make this reviewer not as confident as one should be to recommend publication of this paper in this prestigious journal. The paper is largely anatomical connectivity based but the data and methods as presented fall short.

We are grateful to the reviewer for the appreciation of the value of our work. And we highly appreciate the detailed and helpful comments. We added more anatomical details in the Figures, Methods and Supplementary data set. All the line numbers in the rebuttal letter refer to the revised version of the manuscript with tracked changes. We also uploaded high resolution pdf files for all the figures.

Based on the methods, it appears that only one injection is done in the PVT and BLA. It is important to show what specific areas were transduced as this will have impact on the location of NAc cells transsynaptically transduced. The pattern of these cells does not conform to what is known about projections being studied.

We thank the reviewer for the suggestion. Because a high titer ($>1 \times 10^{13}$ GC/mL) of the virus is needed for the AAV1-Cre based anterograde tagging (Zingg et al., 2017), we used AAV1-Cre virus of $1 \sim 2 \times 10^{13}$ GC/mL for anterograde transsynaptic tagging. This titer of virus resulted in a relatively large infected area in the PVT or BLA, therefore we did only one injection in each region. We made it clear in the Methods part of the revised manuscript (*Line: 340-343, 355-356*).

As the reviewer suggested, previous studies have reported subregions or subpopulations of cells in the PVT or the BLA show different innervation patterns in the NAc. For example, anterior PVT neurons tend to target more dorsal part of the medial NAc shell, while posterior PVT sends dense fiber to the ventromedial NAc (Gao et al., 2020; Li and Kirouac, 2008; Vertes and Hoover, 2008). In the current study, our stereotaxic injections for the PVT were set on AP of -1.4 mm from the bregma which is in the posterior PVT. In spite of the fact that the contaminated thalamic districts amplify a bit past the PVT, those thalamic neurons scarcely send projections to the NAc (Zhu et al., 2016), hence not influencing our conclusions. The stereotaxic injection for the BLA was set on AP of -1.5 mm which covered almost the entire BLA. We illustrated the specific areas that were transduced with AAV1-Cre in **Supplementary Fig. 8 (Rebuttal Fig. 4)**.

Most of the anatomical experiments lack credibility because of the lack of details provided in the methods and the display of the results. The anatomical mappings are of poor quality and do not show the details necessary to judge as a reviewer or potential reader. For example, Fig. 1 needs to show high magnification images of the cells labeled by the PVT and BLA is necessary.

We apologize for the loss of details after file compression. Images were replaced with higher resolution pictures in Fig. 1b, Supplementary Fig. 1a-b and 2. And we added a higher magnification view for Fig. 1b. We also uploaded higher resolution pdf files for all the figures.

Following the reviewer's suggestion, we added details of experiments in the **Methods** (Line: 340-343, 349-365, 381, 437-440), such as the titer of viruses and the volume of injections for each brain location. We also illustrated the injection sites as well as the locations of fiber placement in **Supplementary Fig. 8 (Rebuttal Fig. 4)**. We added a reference atlas to help the reviewers and readers to evaluate the anatomical locations. For example, in **Supplementary Fig. 2 (Rebuttal Fig. 6)**, we added anatomical references for each AP position and replaced images with high magnification pictures in separate channels.

Rebuttal Fig.5 Example image showing labeled NAc^{BLA} (red) and NAc^{PVT} (green) neurons in the double reporter mouse. Scale bar: 100 μ m.

Images are often super saturated and misleading and the patterns of labeling shown in the Supplementary do not appear to follow the boundaries associated the PVT anterograde tracing. Anatomical delineations are not adequate and do not instill confidence. For example, regional cell clusters would be expected in the NAc following targeting of the PVT and BLA.

We thank the review for this suggestion. We replaced pictures in **Supplementary Fig. 2 (Rebuttal Fig. 5)** with high mag images and added anatomical references from 'The Mouse Brain in Seterotaxic Coordinates *Second Edition*' by George Paxinos and Keith B.J. Franklin. We showed that NAc^{PVT} neurons are located in the ventromedial part of NAc as previously reported (Zhu et al., 2016). And regional cell clusters could be observed in the NAc as the reviewer suggested.

Rebuttal Fig.6 Distribution of NAc^{BLA}, NAc^{PVT} neurons at different A-P positions. Distribution of anterogradely labeled neurons in the NAc receiving BLA (NAc^{BLA}: red) or PVT inputs (NAc^{PVT}: green) at different A-P positions. Scale bar: 200 μ m.

More details on the location of the optical fibers in the NAc are necessary considering that the transduced cells throughout the extent of the NAc appears to be occurring.

We thank the reviewer for the suggestion. We illustrated the injection sites as well as the locations of fiber placement in **Supplementary Fig. 8** of the revised manuscript (**Rebuttal Fig. 4**).

The PVT and BLA fibers in the NAc are likely part of a highly collateralized system to project to other areas. Optostimulation of these fibers in the NAc is likely to result in stimulation collaterals to many areas other

than the NAc. Control experiments with fibers place in other regions receiving projections from the PVT and BLA is required for experiments to be conclusive.

We thank the reviewer for raising this important point. Indeed, previous literature has shown that PVT neurons send some axon collaterals to the NAc and the CeA (Dong et al., 2017). To avoid stimulating the PVT or BLA neurons' collaterals, we used AAV1-mediated anterograde transsynaptic tagging to directly stimulate the downstream NAc neurons that received PVT or BLA inputs.

On the other hand, previous literature from our group showed that optogenetic activation of PVT→NAc induces aversion, while the activation of PVT→CeA does not affect the time spent in the RTPP test (Keyes et al., 2020; Zhu et al., 2016). This suggests that the effect of PVT→NAc stimulation is unlikely to arise from collateral projection to CeA. Previous studies from other groups showed that activation of the BLA→NAc pathway is rewarding and the BLA→CeA pathway drives aversion (Namburi et al., 2015), suggesting the effect of BLA→NAc stimulation is unlikely to arise from collateral projection to CeA.

However, following the reviewer's suggestions, we performed the following control experiments. We expressed ChR2 in NAc^{BLA} or NAc^{PVT} neurons, and placed the optic fiber in CeA. In NAc^{BLA}::ChR2 mice, light stimulation of CeA did not affect the preference in the RTPP test and did not support SA. In NAc^{PVT}::ChR2 mice, light stimulation of CeA also had no effect on preference in the RTPP test and did not support SA. These results indicate that, in our NAc^{BLA} and NAc^{PVT} stimulation experiments (**Fig. 2a-f**), the rewarding or aversive behavior effects cannot be attributed to collateral projections to CeA. We added this piece of data to **Supplementary Fig. 5c-e**.

Rebuttal Fig.7. Optical stimulations in the CeA in NAc^{BLA}::ChR2 and NAc^{PVT}::ChR2 mice.

a, Schematic showing viral strategy to transduce NAc^{BLA} or NAc^{PVT} neurons with ChR2 and to place the optic fiber at CeA.

b, Average numbers of nose pokes for NAc^{BLA} (n = 6) and NAc^{PVT} mice (n = 6). Two-way ANOVA: ChR2 x poke ($F_{(1,10)} = 0.2241, P > 0.05$), poke ($F_{(1,10)} = 0.8965, P > 0.05$). Mean \pm s.e.m.

c, Quantification of preference score in real-time place preference test for female NAc^{BLA}::ChR2 (n = 6) and NAc^{PVT}::ChR2 (n = 5) mice. One-sample t-test, not significantly different from theoretical mean 0.

Reviewer #3:

This interesting study from Zhou et al uses an anterograde tracing and optogenetics approach to study the roles of distinct NAc neuron subpopulations in behavior. They find that NAc neurons receiving inputs from BLA are important for positive reinforcement while those receiving inputs from PVT mediate aversive behaviors. The circuit tracing is well-performed and included a nice experiment using Cre/Flp double reporter mice to show that NAcBLA and NAcPVT neurons were (mostly) distinct neuronal populations. The optogenetic experiments were well-controlled and showed that the NAcBLA and NAcPVT subpopulations played different roles in behavior. The authors then built further upon the circuit tracing to show that NAcBLA and NAcPVT subpopulations had different projection profiles and were thus part of two distinct circuits. This is a nice study, and I have only a few important suggestions for edits.

We sincerely appreciate the reviewer's insightful and helpful comments on our manuscript.

1) Only male mice were used in this study. For many reasons, this is not ideal. Going forward, I strongly urge the authors to include female mice in all their experiments. However, I realize some of these experiments involved complex breeding strategies and it would be very time consuming to repeat those experiments for this manuscript. Therefore, I strongly recommend that the authors be transparent throughout the text that their experiments were performed only in males. This should be made clear in the abstract, introduction, and results. The authors should also add a paragraph to the discussion explaining why it is critical for these experiments to be repeated in females.

We thank the reviewer for raising this important point. We agreed that females should be included in all experiments. However, since most experiments involved complex breeding strategies and it would be very time consuming to repeat all experiments. Thus, we choose to repeat the most important experiments in **Fig. 2c and 2f** with female mice. The results showed that the differential effect of NAc^{BLA} and NAc^{PVT} neurons activations in female mice seem to follow the same trend as in male animals (**Supplementary Fig. 6; Rebuttal Fig. 8**). We also make it clear in the **Methods** that for some other experiments only male mice were used (*Line: 330-331, 451-452, 459-460*). All the line numbers in the rebuttal letter refer to the revised version of the manuscript with tracked changes.

Following the reviewer's suggestion, we added a paragraph to discuss the importance of using both male and female animals (*Line: 298-305*).

Rebuttal Fig.8 In female mice, optical stimulation of the NAc^{BLA} and NAc^{PVT} neurons induced reward and aversion, respectively.

a, Average numbers of nose pokes for female NAc::GFP (n = 7), NAc^{BLA}::ChR2 (n = 8) and NAc^{PVT}::ChR2 mice (n = 9). Two-way ANOVA : ChR2 x poke ($F_{(2,21)} = 18.03, P < 0.0001$), poke ($F_{(1,21)} = 16.67, P < 0.001$), ChR2 group ($F_{(2,21)} = 5.751, P < 0.05$), followed by post-hoc Sidak's test. **** $P < 0.0001$. Mean \pm s.e.m.

b, Quantification of preference score in real-time place preference test for female NAc::GFP (n = 8), NAc^{BLA}::ChR2 (n = 7) and NAc^{PVT}::ChR2 (n = 6) mice. One-way ANOVA ($F_{(2,18)} = 15.8, P = 0.0001$) followed by post-hoc Tukey's test. * $P < 0.05$, **** $P < 0.0001$.

2) The discussion is extremely short and mainly rehashes the results. This section needs to be improved. In addition to discussing the need females, the authors should discuss other future directions. One possibility is that they consider whether their behavioral outcomes match the precision of their neural circuit approaches. The approaches used to define and modulate the specific neuronal subpopulations in NAc are state of the art. However, I would argue that the behavioral outcomes used (such as RTTP) rely on less precise summary statistics. Perhaps future experiments should study the microstructure of behavior during behavior. Pose estimation could be useful to the authors considering the precision of their circuit approaches.

We agree with the reviewer that future experiments should benefit from analyzing the microstructure of behaviors and discuss this point in the revised manuscript (*Line: 305-310*). We also reanalyzed the behavior video clips and compared some microstructures of behaviors in the RTTP experiments that were indicative of aversion or stress. This piece of data was added to **Supplementary Fig.7 (Rebuttal Fig.9)** of the revised manuscript.

Rebuttal Fig.9 Microstructures of behaviors in the RTPP test.

- a**, Average numbers of self-grooming during the RTPP test. Mann-Whitney test. ns, not significant.
- b**, Average numbers of rears during the RTPP test. Mann-Whitney test. ns, not significant.
- c**, Numbers of mouse droppings during the RTPP test. Mann-Whitney test. ns, not significant.
- d**, Frequencies of urination during the RTPP test. Mann-Whitney test. ns, not significant.

References

- Dong, X., Li, S., and Kirouac, G.J. (2017). Collateralization of projections from the paraventricular nucleus of the thalamus to the nucleus accumbens, bed nucleus of the stria terminalis, and central nucleus of the amygdala. *Brain Struct Funct* 222, 3927-3943.
- Engelke, D.S., Zhang, X.O., O'Malley, J.J., Fernandez-Leon, J.A., Li, S., Kirouac, G.J., Beierlein, M., and Do-Monte, F.H. (2021). A hypothalamic-thalamostriatal circuit that controls approach-avoidance conflict in rats. *Nat Commun* 12, 2517.
- Gao, C., Leng, Y., Ma, J., Rooke, V., Rodriguez-Gonzalez, S., Ramakrishnan, C., Deisseroth, K., and Penzo, M.A. (2020). Two genetically, anatomically and functionally distinct cell types segregate across anteroposterior axis of paraventricular thalamus. *Nat Neurosci* 23, 217-228.
- Keyes, P.C., Adams, E.L., Chen, Z., Bi, L., Nachtrab, G., Wang, V.J., Tessier-Lavigne, M., Zhu, Y., and Chen, X. (2020). Orchestrating Opiate-Associated Memories in Thalamic Circuits. *Neuron* 107, 1113-1123 e1114.
- Li, S., and Kirouac, G.J. (2008). Projections from the paraventricular nucleus of the thalamus to the forebrain, with special emphasis on the extended amygdala. *J Comp Neurol* 506, 263-287.
- Namburi, P., Beyeler, A., Yorozu, S., Calhoon, G.G., Halbert, S.A., Wichmann, R., Holden, S.S., Mertens, K.L., Anahtar, M., Felix-Ortiz, A.C., *et al.* (2015). A circuit mechanism for differentiating positive and negative associations. *Nature* 520, 675-678.
- Vertes, R.P., and Hoover, W.B. (2008). Projections of the paraventricular and paratenial nuclei of the dorsal midline thalamus in the rat. *J Comp Neurol* 508, 212-237.
- Zhu, Y., Wienecke, C.F., Nachtrab, G., and Chen, X. (2016). A thalamic input to the nucleus accumbens mediates opiate dependence. *Nature* 530, 219-222.
- Zingg, B., Chou, X.L., Zhang, Z.G., Mesik, L., Liang, F., Tao, H.W., and Zhang, L.I. (2017). AAV-Mediated Anterograde Transsynaptic Tagging: Mapping Corticocollicular Input-Defined Neural Pathways for Defense Behaviors. *Neuron* 93, 33-47.

REVIEWERS' COMMENTS

Reviewer #1 (Remarks to the Author):

The authors have addressed my concerns most adequately. I maintain this is a very interesting and important study.

Reviewer #3 (Remarks to the Author):

The authors did a fantastic job responding to the reviews and I support publication.

Reviewer #1 (Remarks to the Author):

The authors have addressed my concerns most adequately. I maintain this is a very interesting and important study.

We are grateful to the reviewer for the appreciation of the value of our work.

Reviewer #3 (Remarks to the Author):

The authors did a fantastic job responding to the reviews and I support publication.

We appreciate the reviewer's recognition of our work.